EMBO
Molecular Medicine

# Dual reporter genetic mouse models of pancreatic cancer identify an epithelial-to-mesenchymal transition-independent metastasis program

Yang Chen[1], Valerie S LeBleu[1], Julienne L Carstens[1], Hikaru Sugimoto[1], Xiaofeng Zheng[1], Shruti Malasi[1], Dieter Saur[2,3] & Raghu Kalluri[1,*]

## Abstract

Epithelial-to-mesenchymal transition (EMT) is a recognized eukaryotic cell differentiation program that is also observed in association with invasive tumors. Partial EMT program in carcinomas imparts cancer cells with mesenchymal-like features and is proposed as essential for metastasis. Precise determination of the frequency of partial EMT program in cancer cells in tumors and its functional role in metastases needs unraveling. Here, we employed mesenchymal cell reporter mice driven by αSMA-Cre and Fsp1-Cre with genetically engineered mice that develop spontaneous pancreatic ductal adenocarcinoma (PDAC) to monitor partial EMT program. Both αSMA- and Fsp1-Cre-mediated partial EMT programs were observed in the primary tumors. The established metastases were primarily composed of cancer cells without evidence for a partial EMT program, as assessed by our fate mapping approach. In contrast, metastatic cancer cells exhibiting a partial EMT program were restricted to isolated single cancer cells or micrometastases (3–5 cancer cells). Collectively, our studies identify large metastatic nodules with preserved epithelial phenotype and potentially unravel a novel metastasis program in PDAC.

**Keywords** dual-recombinase system; metastasis; micrometastasis; pancreatic ductal adenocarcinoma; partial epithelial-to-mesenchymal transition
**Subject Categories** Cancer; Development & Differentiation

## Introduction

Epithelial-to-mesenchymal transition (EMT), the conversion of epithelial cells to mesenchymal phenotype, is associated with the loss of apical–basal polarity and the acquisition of mesenchymal morphology (Hay, 1995; Thiery, 2002; Kalluri & Weinberg, 2009).

Previous studies have collectively offered evidence for the detection of EMT in circulating tumor cells (Rhim et al, 2012; Yu et al, 2013; Javaid et al, 2015) and conducted gain-of-function/loss-of-function assays targeting EMT-inducing transcription factors such as Twist, Snail, Slug, and Zeb1 (Guaita et al, 2002; Arumugam et al, 2009; Wellner et al, 2009; Taube et al, 2010; Tsai et al, 2012). These studies gave mechanistic insights into the molecular basis of EMT and linked the EMT program to metastasis. In the context of pancreatic cancer, lineage tracing studies revealed fibroblast-specific protein-1 (Fsp1, also called S100A4), Zeb1, and Snail-expressing cancer cells, defined as cancer cells exhibiting a partial EMT program, were observed early in tumor formation (Rhim et al, 2012). Furthermore, > 20% of pancreatic cancer cells in macrometastases revealed expression of mesenchymal marker, Fsp1 (Aiello et al, 2016). Using similar genetically engineered mouse models (GEMMs) of pancreatic ductal adenocarcinoma (PDAC), we previously reported that genetic deletion of Snai1 (Snail) or Twist1 (Twist) was dispensable for the formation of metastases but reduced chemoresistance (Zheng et al, 2015). In the context of breast cancer, although genetic deletion of Snai1 reduced metastatic burden (Tran et al, 2014), over-expression of miR-200 (which targets Zeb1 and Zeb2 transcriptional repression of E-cadherin) did not reduce metastasis despite an impact on chemoresistance (Fischer et al, 2015). To capture cancer cells with a partial EMT program, multiple mesenchymal markers have been used for immunolabeling of lineage-tagged cancer cells, which included α-smooth muscle actin (αSMA), Fsp1 (S100A4), Zeb1, and vimentin (Trimboli et al, 2008; Rhim et al, 2012; Fischer et al, 2015; Zheng et al, 2015; Aiello et al, 2016; Zhao et al, 2016). The KPC (LSL-Kras$^{G12D/+}$;Trp53$^{R172H/+}$;Pdx1-Cre) GEMM (Hingorani et al, 2005), combined with the conventional loxP-STOP-loxP-Reporter transgene, has been used for the fate mapping of PDAC cancer cells and their EMT process (Rhim et al, 2012; Zheng et al, 2015). While this approach strictly allows for the reporter labeling of cancer cell based on the Cre-loxP system, the identification of partial EMT program in reporter-labeled cancer cells relies on antibody-based tissue section immunolabeling for mesenchymal

---

1   Department of Cancer Biology, Metastasis Research Center, University of Texas MD Anderson Cancer Center, Houston, TX, USA
2   Department of Medicine II Klinikum rechts der Isar, Technische Universität München, München, Germany
3   German Cancer Research Center (DKFZ) and German Cancer Consortium (DKTK), Heidelberg, Germany
    *Corresponding author. Tel: +1 713 994 5310; E-mail: rkalluri@mdanderson.org

markers that may not accurately capture cancer cells with partial EMT program. Here, we evaluated the frequency of partial EMT program, directly facilitated by dual reporter lineage tracing of metastases associated with pancreatic cancer GEMMs.

Saur and colleagues reported a next-generation dual-recombinase system (DRS) integrating both Cre-loxP and Flippase (Flp)-FRT (Schonhuber et al, 2014). With this system, distinct genes are independently manipulated under the control of Cre and Flp recombinases, respectively. The $FSF$-$Kras^{G12D/+}$;$Trp53^{frt/+}$;$Pdx1$-$Flp$ (KPF) GEMMs exhibit analogous features associated with PDAC progression and metastasis when compared to KPC GEMMs (Schonhuber et al, 2014). Here, we establish multiple transgenic mouse strains using the next-generation DRS and PDAC models to achieve a dynamic dual-fluorescence transition, which allows for monitoring of EMT program in pancreatic cancer cells. Keeping in mind that both fibroblast-specific protein-1 (Fsp1) and α-smooth muscle actin (αSMA) were reported as mesenchymal cell markers associated with EMT in pancreatic cancer (Kalluri & Weinberg, 2009; Wang et al, 2009; Rhim et al, 2012; Zheng et al, 2015; Aiello et al, 2016; Li et al, 2016), we bred the KPF strain with αSMA-Cre or Fsp1-Cre transgenic mice combined with a novel dual-fluorescence-switchable reporter, R26$^{Dual}$ (Rosa26-CAG-loxP-frt-Stop-frt-FireflyLuc-EGFP-loxP-RenillaLuc-tdTomato). This genetic strategy allows Pdx1-lineage cancer cells to express EGFP, while cells that activate αSMA or Fsp1 promoters are positive for tdTomato. Importantly, EGFP$^+$ cancer cells, upon the expression of αSMA or Fsp1, irreversibly lose EGFP expression and gain tdTomato expression. Therefore, during the entire course of their lifespan, cancer cells with an acquired EMT program will remain tdTomato$^+$/EGFP$^-$ even if they subsequently lose αSMA or Fsp1 gene expression, or when potentially reverting to an epithelial morphology upon speculated mesenchymal-to-epithelial transition (MET) to establish metastatic niches. The design of this genetically engineered system enables the lineage tracing of an αSMA- and Fsp1-associated EMT program in spontaneous PDAC tumors and associated metastases.

## Results

### Characterization of EMT with dual-recombinase fluorescence lineage tracing in KPF;αSMA-Cre;R26$^{Dual}$ mice

We first generated the KPF ($FSF$-$Kras^{G12D/+}$;$Trp53^{frt/+}$;$Pdx1$-$Flp$); αSMA-Cre;R26$^{Dual}$ (Rosa26-CAG-loxP-frt-Stop-frt-FireflyLuc-EGFP-loxP-RenillaLuc-tdTomato) mice. The use of αSMA-Cre as a marker for partial EMT transgene reporter was motivated by our previous study, wherein lineage-tracing analyses of PDAC GEMM with conditional loss of Snail or Twist revealed a significant loss of EMT program in tumors, as measured using immunolabeling for αSMA, Zeb1, Zeb2, and Slug (Zheng et al, 2015). Notably, these findings, using KPC;YFP ($LSL$-$Kras^{G12D/+}$;$Trp53^{R172H/+}$;$Pdx1$-$Cre$;$R26^{LSL-YFP}$) mice employing lineage-traced cancer cells (Zheng et al, 2015), were validated using the additional mesenchymal markers such as Fsp1, Zeb1, and vimentin (Appendix Fig S1). The percent of lineage-traced cancer cells expressing Zeb1 or Fsp1 in the primary tumors were similar to the previously reported findings (Aiello et al, 2016). Using multiple anti-αSMA antibodies, including the previously reported antibodies in PDAC GEMMs (Aiello et al, 2016), αSMA-expressing cancer cells with an EMT program could

also be detected in the primary tumors (Appendix Figs S2 and S3). In our current KPF;αSMA-Cre;R26$^{Dual}$ mice, the dual-recombinase system induces oncogenic $Kras^{G12D}$ expression concurrent with heterozygous loss of p53 and an EGFP lineage tracing in Pdx1-Flp-expressing pancreatic epithelial cells. The Flp-FRT-based KPF alleles induced spontaneous PDAC, which exhibited analogous progression and metastasis compared to the KPC model (Fig 1A–C). The PanIN, PDAC, and metastatic lesions revealed prominent expression of cytokeratin-19 (CK19) (Fig 1B and C, Appendix Fig S3). The Pdx1-lineage cancer cells express EGFP, while αSMA-positive myofibroblasts express tdTomato (Fig 1D). EGFP$^+$ cancer cells, upon acquisition of mesenchymal features that include expression of αSMA, irreversibly lose EGFP expression (with the entire EGFP sequence permanently removed by the Cre-loxP mechanism) and express tdTomato. Therefore, the spontaneous EGFP-to-tdTomato fluorescence transition captures the αSMA-related partial EMT program in cancer cells (Fig 1D). The fluorescence switch design ensures that cancer cells with a partial EMT program, once expressing αSMA, will remain tdTomato$^+$/EGFP$^-$ even if/when they revert to epithelial morphology, possibly via mesenchymal-to-epithelial transition (MET).

We evaluated the EGFP-to-tdTomato fluorescence transition driven by αSMA-related partial EMT program in the primary tumors of KPF;αSMA-Cre;R26$^{Dual}$ mice (Fig 2A and B). Pancreatic cancer cells revealed Pdx1-Flp-induced EGFP positivity and the majority of these cells also expressed CK19. In contrast, the αSMA-expressing myofibroblasts (CK19-negative) associated with PDAC desmoplasia revealed tdTomato positivity (Fig 2A and B, Appendix Fig S4A and B). All areas of the (non-necrotic) tumor sections were examined indiscriminately and included both peri-tumoral and intra-tumoral areas. Nearly all CK19$^+$ cancer cells expressed report-driven EGFP (> 95%, Fig 2B). A consistent proportion (~1.8% of cancer cells per visual fields) of CK19$^+$ pancreatic cancer cells were positive for tdTomato (tdTomato$^+$/CK19$^+$), supporting their αSMA-associated launch of a partial EMT program (Fig 2B). To confirm these findings while keeping in mind the heterogeneous levels of expression of CK19, we carried out similar immunolabeling experiments using E-cadherin to capture epithelial cancer cells. Similar results were obtained, wherein E-cadherin labeling captured the majority of EGFP-expressing cancer cells (Appendix Fig S4C). Furthermore, a discrete proportion (0.5%) of EGFP/tdTomato double-positive cells (with diminishing EGFP and emerging tdTomato expression) were also observed (Fig 2B), possibly reflecting retained EGFP proteins despite the start of tdTomato transcription in cancer cells at the onset of the αSMA-Cre-driven partial EMT phenotype.

### Established metastatic tumors reveal an epithelial phenotype without evidence for αSMA-associated partial EMT program

Established metastatic nodules from KPF;αSMA-Cre;R26$^{Dual}$ mice were examined by evaluating 20 different tissue sections per lung and liver of each mouse. All of the examined metastatic cancer cells in the established nodules expressed EGFP without any evidence for tdTomato$^+$ cancer cells (Table 1). The established macrometastases in our study were defined as metastatic nodules containing more than 10 cancer cells (Table 1, Fig 2C and D), as also reported by others (Bailey-Downs et al, 2014). In contrast,

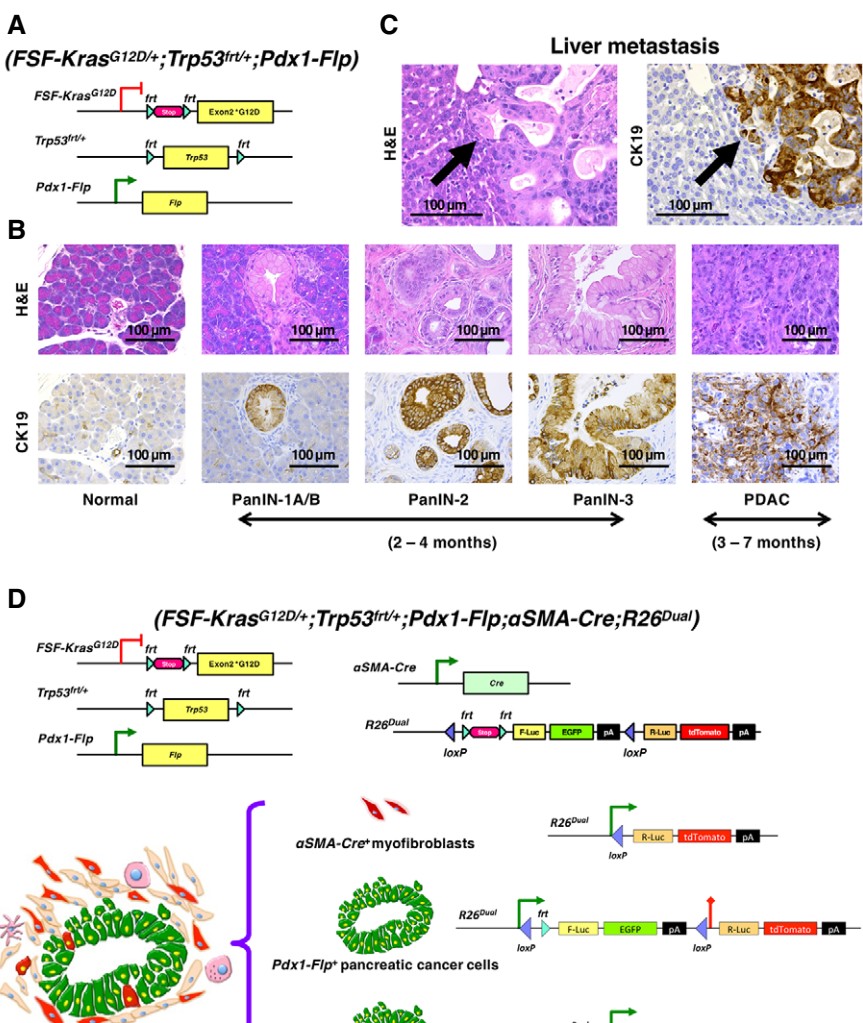

**Figure 1. Dual-recombinase fluorescence lineage tracing of EMT in KPF (FSF-Kras^G12D/+;Trp53^frt/+;Pdx1-Flp);αSMA-Cre;R26^Dual mice.**

A  Genetic strategy to induce oncogenic Kras^G12D and heterozygous p53 loss using the *Pdx1-Flp-FRT* recombination system in KPF mice of the dual-recombinase system (DRS).

B  Representative sections of PanIN (stage 1-3) and PDAC lesions of KPF mice stained by hematoxylin and eosin (H&E) and cytokeratin-19 (CK19) immunohistochemistry. These panels are also depicted in Appendix Fig S3.

C  Representative H&E-stained and CK19-immunostained sections of liver metastases (arrows) in KPF mice. These panels are also depicted in Appendix Fig S3.

D  Genetic strategy to induce EGFP expression in *Pdx1-Flp* lineage and tdTomato expression in *αSMA-Cre* lineage (either myofibroblasts or αSMA-expressing EMT cancer cells) using a novel *Rosa26-CAG-loxP-frt-Stop-frt-FireflyLuc-EGFP-loxP-RenillaLuc-tdTomato* (R26^Dual) tracer.

---

tdTomato^+CK19^+ cancer cells with a partial EMT program were only observed as single cancer cells within the lung and liver parenchyma and/or as part of micrometastases (small clusters of about 3-5 cells) (Table 1, Fig 2C and D). Macrometastases were also shown to lack αSMA expression by IHC (Appendix Fig S3). The EGFP^+ macrometastases without an evidence for an EMT program were also visualized using *ex vivo* imaging (Appendix Fig S5A). Further, the specificity of the dual-recombinase fluorescence lineage-tracing system was examined by immunolabeling tumor sections of KPF;αSMA-Cre;R26^Dual mice using antibodies to αSMA (Aiello *et al*, 2016). The αSMA-Cre-induced tdTomato expression pattern co-localized with anti-αSMA antibody-mediated immunofluorescence labeling in both αSMA-expressing cancer cells and myofibroblasts (Fig 3A and B). Of note, the αSMA-Cre-induced tdTomato expression was noted to also co-localize with Fsp1/S100A4 antibody-based immunolabeling (Fig 3C), another mesenchymal marker reported as important for identifying epithelial cells with an EMT program (Kalluri & Neilson, 2003; Rhim *et al*, 2012; Aiello *et al*, 2016). Collectively, our data suggest that formation of established metastases, without an evidence for an EMT program (CK19^+, no αSMA-Cre-captured partial EMT program), emerges independently from the disseminated cancer cells with features of an EMT program (CK19^+ with αSMA-Cre-captured partial EMT program).

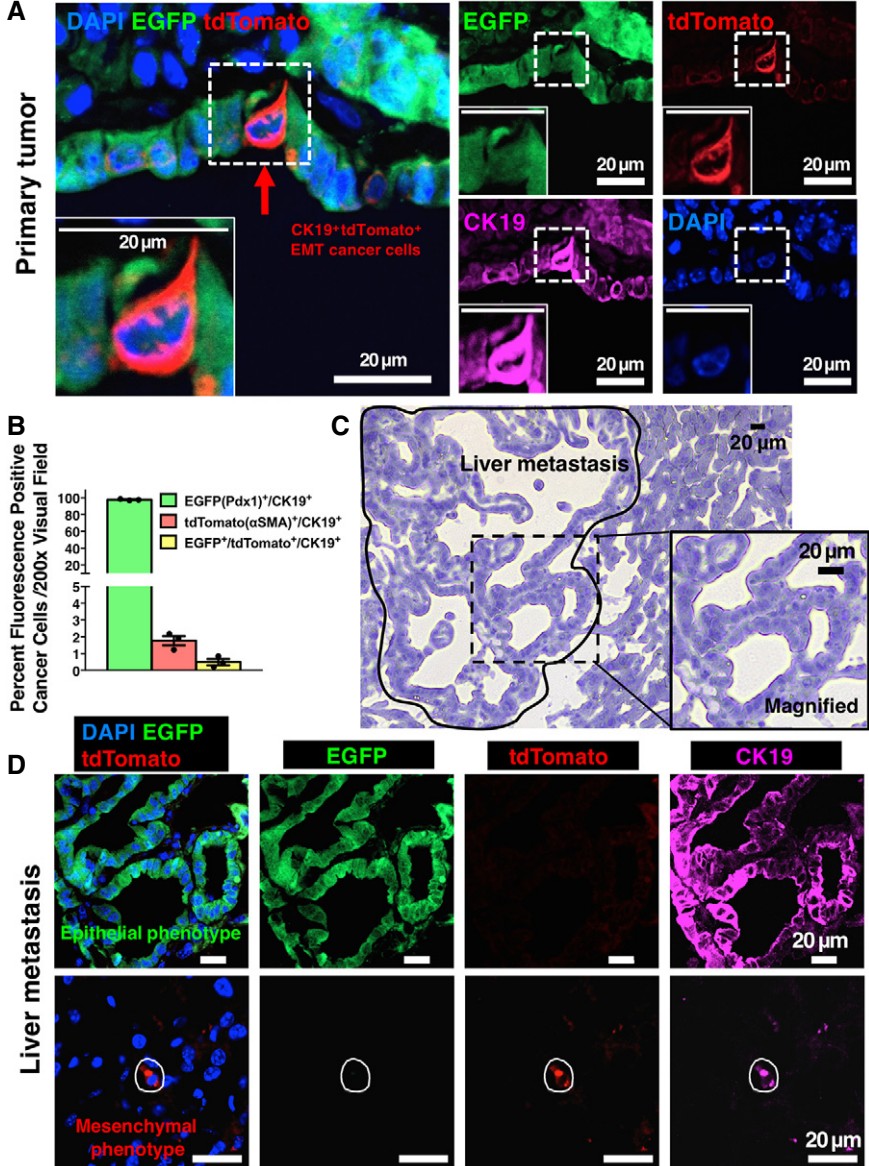

**Figure 2. Established metastases reveal an epithelial phenotype without trace of EMT.**

A    Representative images of primary PDAC tumors from KPF;αSMA-Cre;R26^Dual mice examined for intrinsic EGFP and tdTomato signals, in combination with CK19 immunofluorescence co-staining. Arrow indicates tdTomato⁺CK19⁺ EMT cancer cell in the primary tumor.

B    Quantification of percentage of EGFP-positive, tdTomato-positive, or double-positive cancer cells in primary PDAC tumors (3 visual fields were evaluated per mouse, $n$ = 3 mice; results are presented as mean ± SEM).

C    Representative images of established liver metastasis (circled area) from KPF;αSMA-Cre;R26^Dual mice. Magnified frame shows the histology of part of the metastasis, which is also examined in subsequent immunofluorescence staining.

D    Representative images of liver metastases from KPF;αSMA-Cre;R26^Dual mice examined for intrinsic EGFP and tdTomato signals, in combination with CK19 immunofluorescence co-staining. The circled area indicates a disseminated cancer cell.

Source data are available online for this figure.

---

**Fidelity of the EMT program in PDAC cell lines isolated from KPF; αSMA-Cre;R26^Dual mice**

Based on pancreatic cancer cell expression of EGFP in KPF; αSMA-Cre;R26^Dual mice, we isolated EGFP⁺ primary tumor cells using flow cytometry and further confirmed CK19 expression by fluorescent microscopy (Fig 4A). TGF-β treatment induced epithelial-to-mesenchymal morphological change in these cells, and was associated with EGFP-to-tdTomato transition, further affirming gain in αSMA activity (Fig 4B). TGF-β treatment also upregulated the transcript levels of Fsp1/S100A4, αSMA (*Acta2*), fibronectin (*FN1*), Snail (*Snai1*), Twist (*Twist1*), and type I collagen α1 (*Col1a1*), while suppressing E-cadherin (*Cdh1*) and CK19 (*Krt19*) transcript level (Fig 4C). Although down-regulated by

**Table 1.  Summary of experimental mice in this study.**

| Strain name | | KPF;αSMA-Cre;R26^Dual | | | KPF;αSMA-Cre;R26^mT/mG | | KPF;Fsp1-Cre;R26^Dual | | |
|---|---|---|---|---|---|---|---|---|---|
| Mouse ID | | J37 | J317 | J488 | J402 | A762 | A369 | 219x | 261x |
| DOB | | 15-04-2015 | 24-06-2015 | 05-08-2015 | 13-07-2015 | 08-07-2016 | 18-05-2016 | 20-09-2016 | 20-09-2016 |
| DOD | | 20-08-2015 | 11-09-2015 | 11-06-2016 | 13-12-2015 | 22-10-2016 | 20-08-2016 | 15-12-2016 | 23-01-2017 |
| Age (day) | | 127 | 79 | 311 | 153 | 106 | 94 | 86 | 125 |
| Moribund | | Y | Y | Y | Y | Y | Y | Y | Y |
| Body weight (g) | | NR | 20.48 | 37.46 | 17.21 | NR | NR | 21.5 | 17.1 |
| Tumor weight (g) | | NR | 0.76 | 1.83 | 2.04 | NR | NR | 0.82 | 0.74 |
| Gender | | M | M | M | F | M | M | F | M |
| Metastasis | Total | | | | | | | | |
| Lung Mets | Non-EMT/Macro | 2/8 | Y | Y | – | – | – | – | – | – |
| | EMT/Macro | 0/8 | – | – | – | – | – | – | – | – |
| | Non-EMT/micro | 2/8 | – | – | – | Y | – | Y | – | – |
| | EMT/micro | 3/8 | Y | – | – | Y | – | Y | – | – |
| Liver Mets | Non-EMT/Macro | 5/8 | – | – | Y | Y | Y | Y | Y | – |
| | EMT/Macro | 0/8 | – | – | – | – | – | – | – | – |
| | Non-EMT/micro | 1/8 | – | – | – | Y | – | – | – | – |
| | EMT/micro | 7/8 | – | Y | Y | Y | Y | Y | Y | Y |

ID, identification; DOB, date of birth; DOD, date of death; g, gram; M, male; F, female; Y, yes; NR, not recorded; –, not observed.

*in vitro* TGF-β treatment, CK19 was still highly expressed and easily detectable in cancer cells (Appendix Fig S5B), which enabled the *in vivo* identification of cancer cells with partial EMT phenotype in metastatic sites.

### Dual-fluorescence lineage tracing in KPF;αSMA-Cre;R26^mT/mG mice confirms detection of EMT and non-EMT program-associated metastasis

To further validate our linage tracing system, we next generated the KPF;αSMA-Cre;R26^mT/mG (*Rosa26-CAG-loxP-tdTomato-loxP-EGFP*) mice. All cells in these mice are tdTomato-positive, while αSMA-expressing cells irreversibly lose tdTomato expression and gain EGFP expression (Fig 5A, Appendix Fig S6A), the reverse of the "green-to-red" system reported earlier (Fig 1). Cancer cells with an EMT program (αSMA-expressing cells) will gain EGFP expression. EGFP⁺ pancreatic cancer cells with a partial EMT program (as indicated by CK19⁺ staining) were documented in the primary tumors (Fig 5B, Appendix Fig S6B), consistent with the findings obtained with KPF;αSMA-Cre;R26^Dual mice (Fig 2). Established macrometastases in KPF;αSMA-Cre;R26^mT/mG mice exclusively revealed an epithelial phenotype, with ubiquitous tdTomato expression and without any evidence for EGFP expression (Fig 5C, Table 1). Non-EMT CK19⁺ cancer cells could also be observed as single cells in the lung of KPF;αSMA-Cre;R26^Dual mice (Table 1, Appendix Fig S6C). EGFP-expressing metastatic cancer cells were observed only as single cancer cells or part of micrometastases (colonies of 3–5 cancer cells, Fig 5C and Table 1). This mouse model, harboring the extensively utilized R26^mT/mG reporter transgene, provided additional confirmation to our previous findings.

### Established metastases associated with PDAC reveal an epithelial phenotype without evidence for *Fsp1-Cre*-captured partial EMT program

It has been argued that in the pancreatic cancer, Fsp1/S100A4 but not αSMA is the most consistent and predominant mesenchymal marker associated with the EMT program (Rhim *et al*, 2012; Fischer *et al*, 2015; Aiello *et al*, 2016). In order to address this thesis, we generated the KPF;Fsp1-Cre;R26^Dual mice, wherein the mesenchymal-specific gene promoter Fsp1/S100A4 drives Cre-recombinase expression (Fig 6A, Appendix Fig S7A). In this PDAC lineage-tracing system, EGFP⁺ cancer cells, upon the expression of Fsp1, irreversibly lose EGFP expression and initiate tdTomato expression, generating an EGFP-to-tdTomato fluorescence transition in cancer cells with *Fsp1-Cre*-captured EMT program. Consistent with our previous results (Appendix Fig S1) and results from others (Aiello *et al*, 2016), EGFP-to-tdTomato switch was found in a small portion (~2.7%) of cancer cells in primary tumors (Fig 6B and C). The established metastatic nodules revealed exclusively EGFP positivity (maintained epithelial phenotype) without any detection of *Fsp1-Cre*-captured partial EMT program in the cancer cells (Fig 6D). The metastases were exclusively either EGFP⁺ (established macrometastases) or tdTomato⁺ single cells or micrometastases (colonies of about 3–5 cells) (Table 1), consistent with the findings using KPF;αSMA-Cre;R26^mT/mG and KPF;αSMA-Cre;R26^Dual mice (Table 1). Macrometastases were also shown to lack Fsp1 expression by IHC (Appendix Fig S3). In the primary pancreatic tumor, the Fsp1/tdTomato-positive cancer cells with a partial EMT program were also positive for other mesenchymal markers including αSMA, vimentin, and Zeb1 (Fig 6E, Appendix Fig S7B–D). In contrast, the Fsp1/tdTomato-positive fibroblasts in the primary pancreatic tumor stroma revealed minimal overlap with

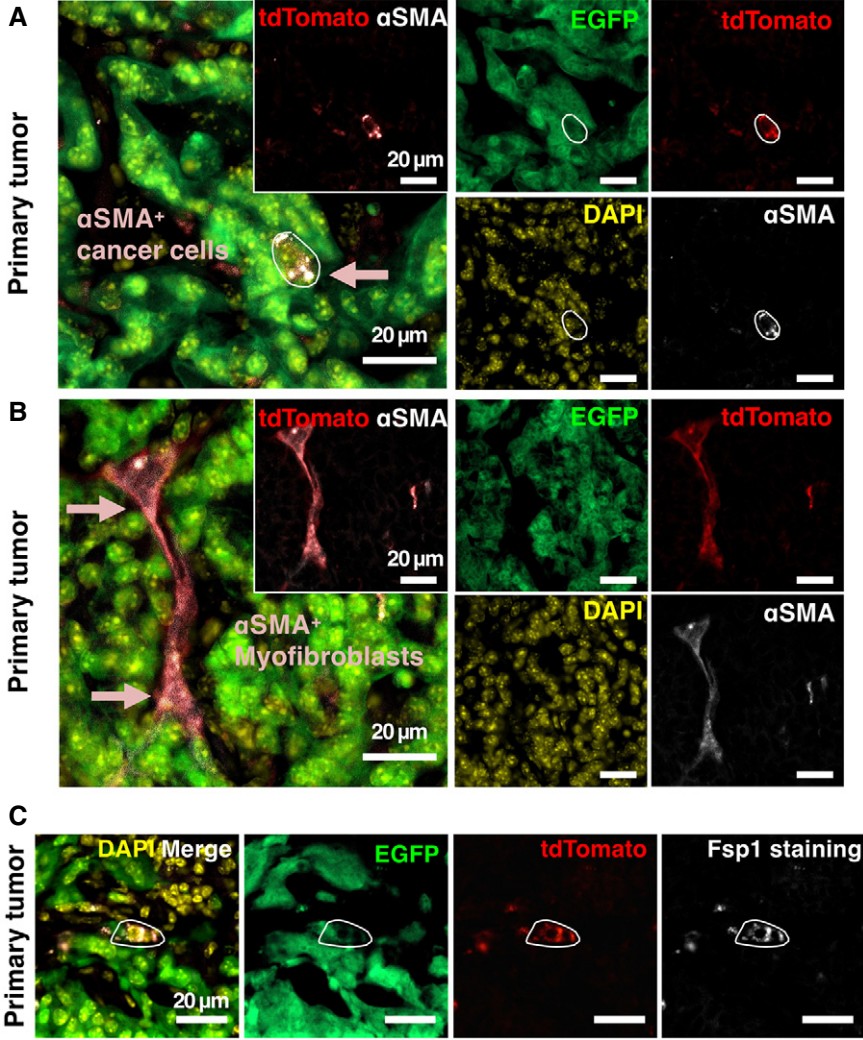

**Figure 3.  Lineage tracing of αSMA-expressing cancer cells and myofibroblasts in KPF;αSMA-Cre;R26^Dual mice.**

A, B  Representative images of colocalization (as indicated by arrows) between αSMA-induced intrinsic tdTomato and αSMA immunofluorescence staining in cancer cells (A) and myofibroblasts (B) of primary tumors from KPF;αSMA-Cre;R26^Dual mice.

C  Representative images of colocalization between αSMA-induced intrinsic tdTomato and Fsp1 immunofluorescence staining in cancer cells of primary tumors from KPF;αSMA-Cre;R26^Dual mice. The circled area indicates tdTomato^+Fsp1^+ EMT cancer cells.

αSMA-expressing fibroblast subpopulation (Appendix Fig S8A). Such minimal colocalization between αSMA and Fsp1 in PDAC stroma was also observed by simultaneous staining with αSMA antibody and Fsp1 antibody on PDAC cryosections of KPF;Cre-negative;R26^Dual mice (Appendix Fig S8B). The recombination efficiency of Fsp1-Cre was confirmed by the robust colocalization between Fsp1-Cre-induced tdTomato and Fsp1 staining in KPF;Fsp1-Cre;R26^Dual tumor sections (Appendix Fig S8C). The Fsp1-Cre-induced tdTomato labeled more cells than Fsp1 antibody staining, supporting the efficiency of our Cre-*loxP*-based lineage-tracing system.

## Discussion

The functional contribution of a partial EMT program in cancer cell dissemination and metastasis remains largely unknown despite extensive investigation. This is in part due to the transient nature of the partial EMT phenotype of cancer cells and the challenges associated with employing fate mapping strategies (Mittal, 2018). Although a partial EMT program was observed in human pancreatic adenocarcinomas in the form of tumor budding (Bronsert *et al*, 2014; Grigore *et al*, 2016), we and others (Rhim *et al*, 2012) also observed cancer cell acquisition of a partial EMT program in PDAC GEMMs in early (PanIN) and late (PDAC) lesions. It remains unclear whether these cancer cells in PanIN with partial EMT program ultimately contribute to the formation of metastases.

We recently reported that PDAC GEMMs lacking Twist or Snail in cancer cells reveal a suppressed EMT program (Zheng *et al*, 2015), and again demonstrated it in this study. Despite the reduction in partial EMT program in PDAC tumors, metastases frequency remained unchanged. The genetic deletion of Zeb1 associated with

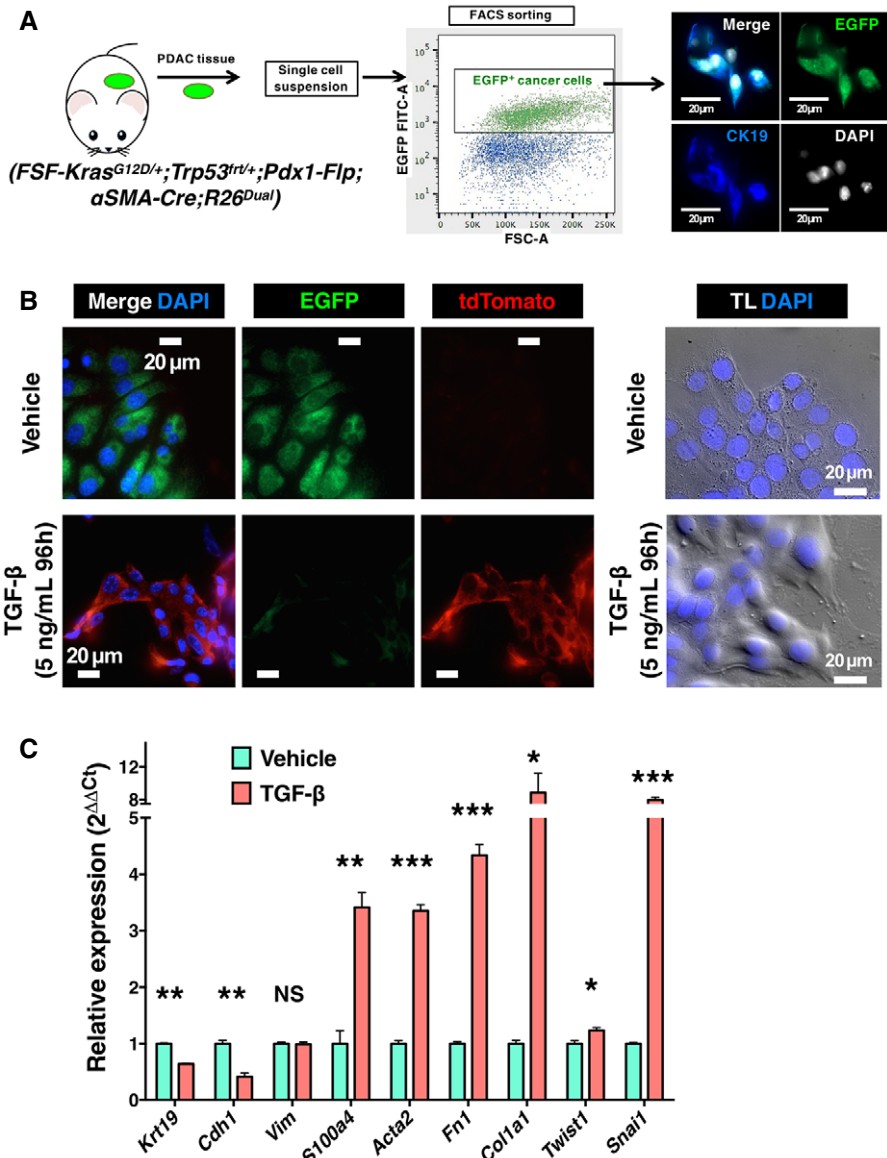

**Figure 4.** *In vitro* induction of EMT using the primary PDAC cells isolated from KPF;αSMA-Cre;R26<sup>Dual</sup> mice.

A    Schematic for the isolation of EGFP-expressing primary pancreatic cancer cells from PDAC tissues of KPF;αSMA-Cre;R26<sup>Dual</sup> mice by FACS.

B    Representative fluorescence and differential interference contrast (DIC) microscopic images of KPF;αSMA-Cre;R26<sup>Dual</sup> primary pancreatic cancer cells treated with TGF-β (5 ng/ml, 96 h) for the *in vitro* induction of EMT.

C    Relative transcript levels of the indicated epithelial or mesenchymal markers in KPF;αSMA-Cre;R26<sup>Dual</sup> primary pancreatic cancer cells with and without TGF-β treatment (5 ng/ml, 48 h), $n = 4$ independent experiments (results are presented as mean ± SEM; Krt19: **$P = 0.0007$, Cdh1: **$P = 0.0041$, Vim: $P = 0.6033$, S100a4: **$P = 0.0082$, Acta2: ***$P = 0.0003$, Fn1: ***$P = 0.0004$, Col1a1: *$P = 0.0424$, Twist1: *$P = 0.0187$, Snai1: ***$P = 0.0003$). Significance determined by paired, two-tailed *t*-test. NS, not significant.

Source data are available online for this figure.

PDAC KPC GEMMs also failed to prevent metastasis in 50 % of mice (Krebs *et al*, 2017). These results support that alternative mechanisms beyond the EMT-MET program may also support the formation of metastatic lesions (Jolly *et al*, 2017). Alternatively, evaluating the role of single transcription factors such as Twist, Snail, and Zeb1 in driving an EMT program and metastasis may insufficiently capture the breadth and complexity of EMT induction in tumors. The definition(s) of the metastasis-permissive partial EMT program is evolving as a result of some unexpected findings (Fischer *et al*, 2015; Zheng *et al*, 2015).

To expand beyond studies that address the requirement of single EMT transcription factors in PDAC metastasis, and to enable visualization of EMT-derived cancer cells in the secondary metastatic tumors, we employed a dual recombinase fate mapping system to lineage trace cancer cells with a partial EMT program. This PDAC GEMM, with the capacity for cellular fate mapping of EMT program

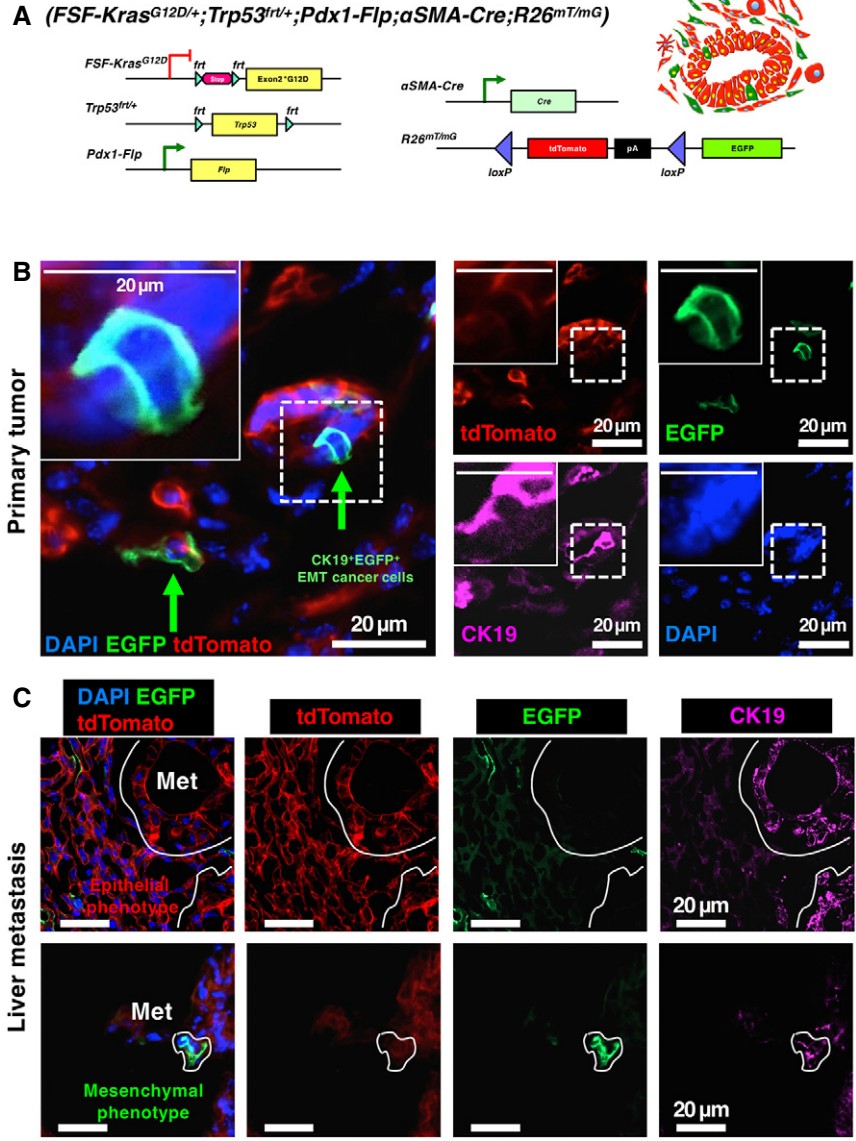

**Figure 5.  Examination of EMT using another fluorescence lineage tracing in KPF;αSMA-Cre;R26^mT/mG mice.**

A   Genetic strategy to induce EGFP expression in *αSMA-Cre* lineage (either myofibroblasts or αSMA-expressing EMT cancer cells) using the *Rosa26-CAG-loxP-tdTomato-loxP-EGFP* (*R26^mT/mG*) tracer.

B, C   Representative images of primary PDAC tumors (B) and liver metastases (Met; C) from KPF;αSMA-Cre;R26^mT/mG mice examined for intrinsic tdTomato and EGFP signals, in combination with CK19 immunofluorescence co-staining. Arrows indicate EGFP⁺CK19⁺ EMT cancer cells in primary tumors. The circled areas (Met) indicate liver metastases.

in cancer cells, offers the potential for novel insights into the mechanism(s) associated with metastasis (Shamir *et al*, 2014; Fischer *et al*, 2015; Zheng *et al*, 2015). In this report, we demonstrate two distinct and mutually exclusive types of metastatic dissemination programs. The cancer cells identified as single cells or micrometastases (small clusters of 3–5 cancer cells) reveal a partial EMT program, using either *αSMA-Cre* or *Fsp1-Cre* as the mesenchymal phenotype tracking system. In contrast, the established, large metastatic nodules consisted exclusively of cancer cells that maintained an epithelial phenotype without an acquisition of *αSMA-Cre*- or *Fsp1-Cre*-captured partial EMT program. Although our current study

employed lineage tracing strategies to support that large metastatic nodules may emerge without a partial EMT program of cancer cells, it is possible that such cells may exert paracrine, fibroblast-like functions and contribute to the formation of large metastatic nodules.

The larger size of non-EMT metastases suggests a potentially proliferative feature of such metastatic nodules. This would certainly align with the notion that a proliferative program is likely suspended in the cancer cells with an EMT program (Ozdemir *et al*, 2014; Zheng *et al*, 2015). Further, a recent study highlighted that epithelial-like breast cancer cells expressing Epcam were more proliferative than those that did not express Epcam and were

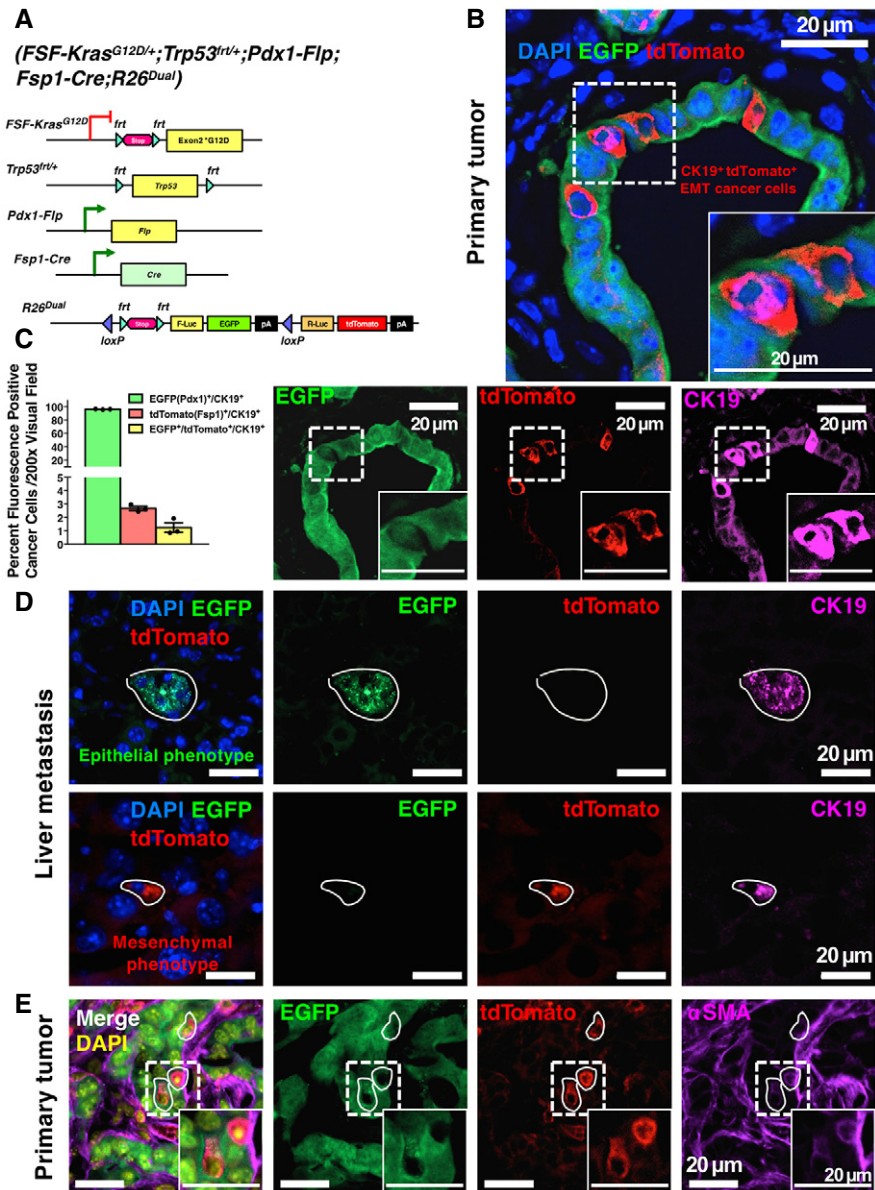

**Figure 6. Indication of EMT with Fsp1-driven fluorescence-switching lineage tracing in KPF;Fsp1-Cre;R26^Dual mice.**

A   Genetic strategy to induce EGFP expression in *Pdx1-Flp* lineage and tdTomato expression in *Fsp1-Cre* lineage (either fibroblasts or Fsp1-expressing EMT cancer cells) in KPF;Fsp1-Cre;R26^Dual mice.

B   Representative images of primary PDAC tumors from KPF;Fsp1-Cre;R26^Dual mice examined for intrinsic tdTomato and EGFP signals, in combination with CK19 immunofluorescence co-staining.

C   Quantification of percentage of EGFP-positive, tdTomato-positive, or double-positive cancer cells in primary tumors (3 visual fields were evaluated per mouse, *n* = 3 mice; results are presented as mean ± SEM).

D   Representative images of metastases from KPF;Fsp1-Cre;R26^Dual mice examined for intrinsic tdTomato and EGFP signals, with CK19 immunofluorescence co-staining. The circled areas indicate liver metastases.

E   Representative images of colocalization (as indicated by circled areas) between Fsp1-induced intrinsic tdTomato and αSMA immunofluorescence staining in cancer cells of primary tumors from KPF;Fsp1-Cre;R26^Dual mice.

Source data are available online for this figure.

classified as subpopulations of cancer cells with EMT program (Pastushenko *et al*, 2018). Our study also suggests that the cancer cells with a partial EMT program, as noted in the micrometastases, may reflect latent, and perhaps more therapeutically recalcitrant

metastatic cancer cells (Shah *et al*, 2007; Yin *et al*, 2007; Arumugam *et al*, 2009; Wang *et al*, 2009; Hugo *et al*, 2016). It is intriguing to propose that such rare, perhaps more latent metastatic cancer cells that arose via acquisition of a partial EMT program reflect cancer

stem cell-like phenotype associated with pancreatic cancer (Valle *et al*, 2018), although this proposition needs experimental evidence.

Our studies clearly demonstrate that the overlap between the αSMA and Fsp1/S100A4 is minimal in the stromal fibroblasts but is significantly high in the cancer cells with a partial EMT program. Our results suggest that the *αSMA-Cre-* and *Fsp1-Cre-*associated EMT program likely capture, at least in part, a common subset of cancer cells. Additionally, this study supports previous studies that cancer cells associated with an *αSMA-* and *Fsp1-Cre*-driven EMT program are rare events in the pancreatic tumors. Our study cannot exclude the possibility that mesenchymal markers, other than αSMA and Fsp1, may identify distinct EMT program(s); however, some reports have suggested that Fsp1 is the predominant mesenchymal marker (with > 20% positivity) in PDAC-associated EMT program (Rhim *et al*, 2012). Collectively, our findings offer novel insights into the EMT-independent mechanism(s) associated with PDAC metastasis.

# Materials and Methods

## Mice

*FSF-Kras*$^{G12D/+}$ (Schonhuber *et al*, 2014), *Pdx1-Flp* (Schonhuber *et al*, 2014), *Trp53*$^{frt/+}$ (Lee *et al*, 2012), *LSL-Kras*$^{G12D/+}$ (Hingorani *et al*, 2005), *Pdx1-Cre* (Hingorani *et al*, 2005), *αSMA-Cre* (LeBleu *et al*, 2013), and *Fsp1-Cre* (Xue *et al*, 2003; Bhowmick *et al*, 2004) mouse strains were previously documented. *Snail*$^{loxP/loxP}$ mice were kindly provided by S. J. Weiss. *Twist*$^{loxP/loxP}$ mice were kindly provided by R. R. Behringer via the Mutant Mouse Regional Resource Center (MMRRC) repository. Characterization of genotyping and disease phenotypes for the *FSF-Kras*$^{G12D/+}$;*Trp53*$^{frt/+}$; *Pdx1-Flp* (referred to as KPF) mice was performed as previously described (Schonhuber *et al*, 2014). The KPF mice were bred to *αSMA-Cre*, *Fsp1-Cre*, *Rosa26-CAG-loxP-tdTomato-loxP-EGFP* (referred to as R26$^{mT/mG}$), or *Rosa26-CAG-loxP-frt-Stop-frt-FireflyLuc-EGFP-loxP-RenillaLuc-tdTomato* (referred to as R26$^{Dual}$). The resulting progeny, termed as KPF;αSMA-Cre;R26$^{Dual}$ (*n* = 3), KPF;αSMA-Cre; R26$^{mT/mG}$ (*n* = 2), and KPF;Fsp1-Cre;R26$^{Dual}$ (*n* = 3), were maintained on a mixed *C57/Bl6;129S6/SvEv;BALB/c* genetic background. Both males and females were used indiscriminately. These mice used in this study develop invasive PDAC at the age of 3–7 months and were sacrificed for tissue collection at the age of 3–10 months (Table 1). The aforementioned experimental mice with desired genotypes were monitored and analyzed with no randomization or blinding. R26$^{mT/mG}$ mice were purchased from Jackson Laboratories (Stock No. 007676). The novel R26$^{Dual}$ reporter allele was constructed based on a modified design similar to that of R26$^{mT/mG}$ reporter allele, with an additional *Frt-Stop-Frt* element controlling the EGFP expression. All mice were housed under standard housing conditions at MD Anderson Cancer Center (MDACC) animal facilities, and all animal procedures were reviewed and approved by the MDACC Institutional Animal Care and Use Committee.

## Histology and immunohistochemistry

For paraffin-fixed samples, mouse tissues were fixed in 10% neutral buffered formalin, embedded in paraffin, and sectioned at 5 μm thickness. In search for metastases, 20 sections for each tissue type (lung or liver) per mouse were randomly selected from the serial sectioning (with 100 μm interval) of the entire lung or liver lobe. Sections were processed for hematoxylin and eosin (H&E) staining. Microscopic metastases were examined in H&E-stained tissue sections of the liver and lung. Images were captured with a Leica DM 1000 LED microscope and an MC120 HD Microscope Camera with Las V4.4 Software (Leica). Formalin-fixed, paraffin-embedded sections were processed for immunohistochemical staining as previously documented (Zheng *et al*, 2015). Sections were incubated with primary antibodies: αSMA (M0851, Dako, 1:100), CK19 (ab52625, Abcam, 1:200), or Fsp1 antibody (A5114, Dako, 1:100), then biotinylated secondary antibodies, and streptavidin HRP (Biocare Medical). For all immunolabeling experiments, sections were developed by DAB and counterstained with hematoxylin.

## EGFP/tdTomato visualization and immunofluorescence

Tissues from those strains with R26$^{Dual}$ or R26$^{mT/mG}$ lineage tracing, expressing intrinsic EGFP and tdTomato, were fixed in 4% paraformaldehyde overnight at 4°C and equilibrated in 30% sucrose overnight at 4°C. Tissues were then embedded in O.C.T. compound (TissueTek) and processed for 5-μm-thick cryosections. In search for metastases, 20 cryosections for each tissue type (lung or liver) per mouse were randomly selected from the serial sectioning (with 100-μm interval) of entire lung or liver lobe. Sections were blocked for 1 h with 4% cold water fish gelatin (Aurion) and immune-stained overnight at 4°C with αSMA antibody (M0851, Dako, 1:100), CK19 antibody (ab52625, Abcam, 1:200), E-cadherin (3195, Cell Signaling, 1:200), Fsp1 antibody (A5114, Dako, 1:100), vimentin (ab45939, Abcam, 1:1,000), or Zeb1 (Novus Biologicals, NBP1-05987, 1:100)., followed by incubation with AlexaFluor647 secondary antibodies (Invitrogen). Staining for αSMA (M0851, Dako, 1:100) was performed with Mouse-on-Mouse (M.O.M.) kit (Vector Laboratories) following the manufacturer's instructions. Slides were then mounted with DAPI-containing Vectashield Mounting Medium (Vector Laboratories), visualized under the LSM800 confocal laser scanning microscope, and analyzed with ZEN software (Zeiss). Fluorescence microscopic images were obtained using constant settings (laser power, filters, gain, and offset, see accompanying source data). The imaging software automatically adjusted brightness (image display cutoff value) to account for overexposure (if any). Both image capture and image display setting are listed in the accompanying source data. Pseudocolored images were generated in order to represent the indicated fluorescence signals. The EGFP-positive, tdTomato-positive, or double-positive cancer cell subpopulations in primary PDAC tumors were quantified and expressed as the percent fluorescence-positive cells among total cancer cells per visual field. The total cancer cell number was counted based on CK19 and DAPI immunolabeling. Established macrometastases are defined as clusters containing more than 10 metastatic cells. Micrometastases are defined as solitary cells or small clusters with < 10 metastatic cells (generally clusters of 3–5 cells). Alternatively, 5-μm-thick formalin-fixed, paraffin-embedded sections were processed for immunofluorescence staining using the following primary antibodies: αSMA mouse antibody (M0851, Dako, 1:100), αSMA rabbit antibody (ab5694, Abcam, 1:100), and GFP (ab13970, Abcam, 1:400). Secondary antibodies conjugated to Alexa Fluor 488 and/or 594 were used following primary antibody incubation. For the quantification

of EMT-positive cells in the context of *Snai1* or *Twist1* cancer cell-specific loss, 5-μm-thick formalin-fixed, paraffin-embedded sections were deparaffinized and serially labeled for Fsp1 (A5114, DAKO, 1:6,000), Zeb1 (NBP1-05987, Novus, 1:500), or vimentin (CS5741, Cell Signaling, 1:200) using Super-Picture polymer broad spectrum-HRP-conjugated secondary antibody (878963, ThermoFisher) and TSA-FITC (NEL760001KT, Perkin Elmer, 1:50), and YFP (anti-GFP, ab13970, Abcam, 1:2,000) using an anti-chicken-HRP-conjugated secondary antibody (613120, Invitrogen, 1:1,000) and TSA-Cy3 (NEL760001KT, Perkin Elmer, 1:100). Five representative 400× images (Zeiss Observer.Z1) were quantified for the percent double-positive cells out of total number of cells in the image (inForm Advanced Image Analysis Software, Perkin Elmer).

### Isolation of primary pancreatic adenocarcinoma cells and myofibroblasts from PDAC tissues

Isolation of primary PDAC cell line was performed as previously described with minor modifications (Zheng *et al*, 2015). Fresh PDAC tissues from KPF;αSMA-Cre;R26$^{Dual}$ mice were minced with sterilized lancets, digested with collagenase IV (17104019, Gibco, 4 mg/ml)/dispase II (17105041, Gibco, 4 mg/ml)/RPMI at 37°C for 0.5 h, filtered by 70-μm cell strainers, resuspended in RPMI/20% FBS, and seeded into type I collagen-coated dishes (354401, Corning). Cells were cultured in RPMI medium containing 20% FBS and 1% penicillin–streptomycin–amphotericin B (PSA) antibiotic mixture. Pdx1-driven EGFP-positive cancer cells were further sorted by FACS (BD FACSAria™ II sorter; South Campus Flow Cytometry Core Laboratory of MD Anderson Cancer Center) based on EGFP signals. All studies were performed on cells cultivated for < 20 passages. For the *in vitro* induction of EMT, KPF;αSMA-Cre;R26$^{Dual}$ primary PDAC cells cultured within Lab-Tek Chamber Slides (ThermoFisher/Nunc) were treated with either 5 ng/ml recombinant TGF-β (R&D Systems) or vehicle (4 mM HCl in H$_2$O with 1 mg/ml BSA) in 1% FBS medium for 48–96 h. RNA from vehicle or TGF-β-treated cells was extracted with RNeasy RNA Mini Kit (QIAGEN), used for cDNA synthesis using Reverse Transcription Kit (Applied Biosystems), and subjected to the qRT–PCR using SYBR Green Master Mix (Applied Biosystems). The expression level of indicated genes was normalized to the expression of Gapdh as housekeeping gene. The relative expression is presented as fold change ($2^{ΔΔCt}$) with the control group set as a value of 1. qRT–PCR primers are as follows: CK19 (*Krt19*) F 5′-TCCCAGCTCAGCATGAAAGCT-3′; CK19 R 5′-AAAACCGCTGATCACGCTCTG-3′; E-cadherin (*Cdh1*) F 5′-AACTGCATGAAGGCGGGAAT-3′; E-cadherin R 5′-TTTCGAGTC ACTTCCGGTCG-3′; vimentin (*Vim*) F 5′-CTTGAACGGAAAGTGGA ATCCT-3′; vimentin R 5′-GTCAGGCTTGGAAACGTCC-3′; Fsp1 (*S100A4*) F 5′-TTGTGTCCACCTTCCACA-3′; Fsp1 R 5′-GCTGTC CAAGTTGCTCAT-3′; αSMA (*Acta2*) F 5′-GTCCCAGACATCAGGGA GTAA-3′; αSMA R 5′-TCGGATACTTCAGCGTCAGGA-3′; fibronectin (*FN1*) F 5′-GCTCAGCAAATCGTGCAGC-3′; fibronectin R 5′-CTAGG TAGGTCCGTTCCCACT-3′; Col1a1 F 5′-CTCCTCTTAGGGGCCACT-3′; Col1a1 R 5′-CCACGTCTCACCATTGGGG-3′; Twist (*Twist1*) F 5′-CTGCCCTCGGACAAGCTGAG-3′; Twist R 5′-CTAGTGGGACGCGGA CATGG-3′; Snail (*Snai1*) F 5′-CACACGCTGCCTTGTGTCT-3′; Snail R 5′-GGTCAGCAAAAGCACGGTT-3′; Gapdh F 5′-AGGTCGGTGTGAA CGGATTTG-3′; Gapdh R 5′ TGTAGACCATGTAGTTGAGGTCA-3′. Statistical analyses were performed on ΔCt.

**The paper explained**

**Problem**
Epithelial-to-mesenchymal transition (EMT), a well-recognized eukaryotic cell differentiation program during embryonic development, is also observed in invasive tumors. Precise identification of partial EMT program in primary tumors and metastases of pancreatic ductal adenocarcinoma (PDAC) remains challenging. This is partly due to the transient nature of the partial EMT phenotype of cancer cells and the challenge in lineage-tracing the EMT program.

**Results**
This study utilizes the dual-recombinase system-driven genetic mouse model of spontaneous pancreatic cancer in combination with fluorescence-switching reporters and mesenchymal fate mapping transgenes (αSMA-Cre and Fsp1-Cre), to lineage trace EMT program. We observed both αSMA- and Fsp1-Cre-mediated partial EMT programs in the primary PDAC tumors, as captured by our lineage-tracing models. The established lung and liver metastases were predominantly composed of cancer cells without evidence for partial EMT program. Metastatic cancer cells with a partial EMT phenotype were observed only as disseminated single cancer cells or micrometastases (3–5 cells). The two classes of disseminated cancer cells were strictly exclusive from one another.

**Impact**
Our novel lineage-tracing system identifies established macrometastases with a preserved epithelial phenotype and without evidence for partial EMT program. This study provides new insights into metastasis that can emerge independent of partial EMT program in pancreatic cancer.

### Statistics

Statistical analyses were performed with paired, two-tailed *t*-test or one-way ANOVA with Tukey's multiple comparisons test using GraphPad Prism (GraphPad Software, San Diego, CA), as indicated in the figure legends. A *P*-value < 0.05 was considered statistically significant. Exact *P*-values are listed in the figure legends. Error bars depicts the standard error of the mean (SEM) when multiple visual fields were averaged to produce a single value for each animal which was then averaged again to represent the mean bar for the group in each graph.

# Data availability

All source data were included with the manuscript.

Expanded View for this article is available online.

# Acknowledgements

We wish to thank S. Yang for designing and generating the schematic depiction of the lineage-tracing strategies, and K. M. Ramirez and R. Jewell at the South Campus Flow Cytometry Core Laboratory of MD Anderson Cancer Center for flow cytometry cell sorting and analyses (in part supported by NCI P30CA16672). This work was primarily supported by the Cancer Prevention and Research Institute of Texas (CPRIT). Research in the Kalluri laboratory is also supported by NCI PO1CA117969 and CPRIT Award RP150231. The LeBleu laboratory is supported by UT MDACC Khalifa Bin Zayed Al Nahya Foundation, and MDACC Small Animal Imaging Facility is supported by NIH P30-CA016672 and 5U24-CA126577.

## Author contributions

RK conceptually designed the strategy for this study, provided intellectual input, and contributed to writing the manuscript. DS contributed to the establishment of the dual-recombinase system mouse models. VSL helped design experimental strategy, provided intellectual input, and edited the manuscript and figure presentation. YC generated the genetically engineered mouse models of PDAC, performed the experiments, collected the tissues for analysis, performed the analyses, composed the figures, and wrote the manuscript. JLC performed immunostaining and quantification presented in Appendix Fig S1, edited the manuscript, and provided advice. HS assisted with experiments, edited the manuscript, and offered experimental advice. XZ participated in the generation of GEMMs for the data presented in Appendix Fig S1, edited the manuscript, and provided advice. SM provided technical support for experiments that included genotyping analyses and tissue collection/sectioning.

## Conflict of interest

The authors declare that they have no conflict of interest.

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
