## [Review Process File · EMBO Molecular Medicine]

Dual reporter genetic mouse models of pancreatic cancer identify an epithelial to mesenchymal transition independent metastasis program

Yang Chen, Valerie S. LeBleu, Julienne L. Carstens, Hikaru Sugimoto, Xiaofeng Zheng, Shruti Malasi, Dieter Saur and Raghu Kalluri

Review timeline:

Submission date:	07 March 2018
Editorial Decision:	03 April 2018
Revision received:	01 June 2018
Editorial Decision:	06 July 2018
Revision received:	12 July 2018
Accepted:	23 July 2018

Editor: Lise Roth

Transaction Report:

1st Editorial Decision

03 April 2018

Thank you for the submission of your manuscript to EMBO Molecular Medicine. We have now heard back from the three referees whom we asked to evaluate your manuscript.

As you will see from the reports below, while referees 1 and 2 are overall positive and support publication of the article in EMBO Molecular Medicine (pending appropriate revisions), referee 3 is skeptical regarding the lineage tracing experiments and the conclusions of the study. Therefore, strengthening of the data to fully support the conclusions (number of mice, of imaging fields, additional controls, markers characterization, thorough presentation and discussion of the results) and addressing the reviewers concerns in full will be necessary for further considering the manuscript in our journal. EMBO Molecular Medicine encourages a single round of revision only and therefore, acceptance or rejection of the manuscript will depend on the completeness of your responses included in the next, final version of the manuscript.

Please also contact us as soon as possible if similar work is published elsewhere. If other work is published, we may not be able to extend the revision period beyond three months.

I look forward to receiving your revised manuscript.

***** Reviewer's comments *****

Referee #1 (Remarks for Author):

The study of Chen et al. describes results of dual reporter lineage tracing of metastasis forming cells in pancreatic cancer (PDAC). Previously the same group has provided results that challenged the concept of EMT-mediated metastasis by showing that PDAC metastases formation does not depend on the EMT transcription factors Snail or Twist. Here, to better trace the transient events of EMT potentially contributing to metastasis, the authors used a genetically engineered mouse models, where the epithelial cancer cells irreversibly lose EGFP expression and gain tdTomato expression upon the expression of the EMT markers α SMA or Fsp1. The authors conclude that the metastatic lesions exhibit either an exclusive epithelial or partial EMT phenotype, whereas the so-called partial EMT-phenotype is limited to individual cells and micrometastasis, but excluded from macrometastatic lesions. To confirm this still provocative result, they use another mouse model, where only the cells that induce α SMA lose tdTomato and gain EGFP.

The study is interesting, clearly presented and provides further evidence for the dispensability of EMT for the metastasis formation.

Specific comments:

1. Figure 1 nicely shows the PDAC progression in the used model by immunohistochemistry (ihc). An informative addition would be ihc characterization of both the primary tumors and metastatic tissues with CK19 as well as α SMA & FSP1.
2. Figure 2A and 6C show even stronger CK19 in the α SMA+ and FSP1+ cells with gained tdTomato, compared to the neighboring epithelial cancer cells, whereas in Figure 5B CK19 is faint in the FSP1+ cancer cell with gained EGFP in the area of otherwise strongly CK19+ epithelial cancer cells: How specific is CK19 detection?
3. Is CK19 expression retained upon full EMT in the PDAC cells? The authors should show CK19 mRNA and protein expression changes upon TGF- β -induced EMT in Figure 4 B and C. Do the vimentin+/Zeb1+/FSP1+ and YFP+ cells express CK19 in Figure S1A-C? Since the detection of cancer cells rely on CK19, this should be clarified.
4. The data shows that the macroscopic metastatic lesions have been formed primarily of cells that have not experienced EMT, but after undergoing EMT could some mesenchymal cancer cells contribute to the fibroblast population? It should be discussed, what happens to the cells in the mesenchymal cancer cell-positive microlesions; don't they proliferate at all?
5. On page 9, the authors refer to Figure S4C to indicate the existence of EGFP+ α SMA-expressing metastatic cancer cells as single cells or micrometastasis, but this figure only shows EGFP+ myofibroblasts and EGFP- cancer cells.
6. Figure S3B lacks the red signal mark in the single channel image (tdTomato).

Referee #2 (Remarks for Author):

In their Manuscript, Chen et al provide evidence for an EMT independent metastatic process in pancreatic cancer. By employing a Dual-Recombinase System and mesenchymal cell reporters (α SMA-Cre and Fsp1-Cre) within genetically engineered mice, they show that partial EMT program is restricted to isolated single cancer cells building micrometastases (3-5 cancer cells), while established macrometastases preserve a purely epithelial phenotype, indicating two distinct and mutually exclusive types of metastatic dissemination programs.

This is a well-designed study and the authors are to be commended for their deep investigation. The manuscript is also well written and the topic is certainly of interest.

Considering extensive focus on the concept of EMT, I would expect some more in-depth discussion on the properties of cells with partial EMT, apart from their ability to form micrometastases, for example stem cell-like properties and role in chemoresistance.

Also, some reference in studies of human PDAC material reporting a similar phenomenon (in form of tumor budding) would help to further understand the contribution of partial EMT program in cancer cell dissemination.

In a recently published study (Krebs et al, Nat Cell Biol. 2017 May;19(5):518-529), another group has shown that in the same pancreatic cancer model, driven by Pdx1-cre-mediated activation of mutant Kras and p53 (KPC model), the EMT-TF Zeb1 is a key factor for the formation of precursor lesions, invasion and metastasis. The authors could perhaps more extensively comment on the possibility of different EMT transcription factors being implicated in driving metastasis.

Referee #3 (Comments on Novelty/Model System for Author):

The lineage tracing models used by the authors do not adequately address the problem, they rather reproduce the uncertainties and problems of previously published lineage tracing experiments. The results thus are not adding novel insights to the current debate on EMT and metastasis.

Referee #3 (Remarks for Author):

The manuscript submitted by Chen et al. reports results that raise the possibility of an EMT-dependent micrometastasis program and a EMT-independent macrometastasis program in the well-established KP metastatic mouse model of PDAC. The authors used a genetic dual recombinase-based and a more conventional Cre-reporter-based lineage tracing approach to genetically mark cancer cells that have undergone an EMT. Here, they also demonstrate that the KP mouse model can be driven by PDX-Flippase and Frt-flanked alleles of activated Ras and p53 to induce carcinogenesis. EMT marker driven Cre expression is then used to induce a color switch in cancer cells undergoing an EMT from GFP to dTomato. However, in this system, mesenchymal cells of the tumor stroma will also gain expression of dTomato. Cancer cells with an EMT program could be found in the primary tumor and as single disseminated cells or micrometastasis in lung and liver. In contrast, established macrometastasis consisted of cells that never had activated an EMT. Based on these findings, the authors conclude that two distinct and mutually exclusive types of metastatic dissemination programs have been functional and they suggest the existence of a novel EMT-independent metastasis program.

The authors' observations are certainly interesting. However, the results should be seen in the context of the current debate on the use of certain EMT markers and Cre-driver lines to actually detect and visualize cancer cells undergoing an EMT *in vivo*. Here, the manuscript does not provide additional new insights.

First of all, the authors use SMA-Cre driver line to visualize cancer cells that have undergone an EMT. A recent comment by Aiello et al. (2017) in Nature shows that SMA is "an unreliable marker of EMT in the KPCY mouse model" and they question the suitability of this line for the adequate detection of an EMT. The authors here have used in addition a Fsp1-Cre driver line to visualize cancer cells that have undergone an EMT. While Aiello et al. demonstrate that Fsp1 is expressed in >20% of the cells in a KPC tumor, here only 2.7% of cells are detected. Based on expression of Fsp1, Zeb1 and E-cadherin, Rhim et al. (Cell, 2012) report that approximately 42% of cancer cells in the KPC mouse models have undergone an EMT, meaning EMT would not be a rare event.

In addition, the expression of SMA and FSP1 do not substantially overlap questioning the mesenchymal cell-specific expression of these markers. Along these lines, in the Introduction the authors cite the reports by Aiello et al. (2016 and 2017) that Fsp1-positive cells are found in macrometastases in the KPC model. However, it is not addressed later on why the lineage tracing does not recapitulate this observation.

A major drawback of the lineage tracing experiments used here is that not only cancer cells can gain dTomato expression upon Cre-mediated recombination but also all mesenchymal cells in the tumor. This fact makes the analysis rather ambiguous, as can also be seen by the rather poor quality of marker co-staining shown throughout the manuscript. Higher resolution and three-dimensional reconstructions are required to obtain data which convincingly support the authors' conclusions.

Finally, the lineage tracing experiments are somehow leading themselves "ad absurdum" by using the epithelial marker CK19 to detect cancer cells. The authors escape this problem in stating that dTomato+/CK19+ cells are in a "partial EMT". However, no further characterization is offered to support this notion, also to characterize the single EMT cancer cells in micrometastasis in lung and liver. In the past, the use of cytokeratins has been a major roadblock in studying EMT in cancer progression, because it may completely ignore any EMT cells.

Since the levels of EMT cancer cells detected by dTomato and by CK19-expression are rather low at ~2 cells per field in a highly aggressive primary tumor, one wonders whether these cells are really EMT cells. They should be further characterized to prove their identity. Is CK19 expression retained during the EMT processes marked by α SMA or Fsp1? Are CK19 positive cells with an EMT program also E-cadherin positive? The authors should comment on this.

Specific comments:

1. Importantly, EMT-independent mechanisms of tumor dissemination have been well described. Hence, the report of a possible EMT-independent metastasis program is not novel.
2. The authors observed cells that have undergone an EMT as single disseminated cells or in micrometastatic lesions exclusively. Were these lesions exclusively single or also multi-colored? It is possible that epithelial and mesenchymal cancer cells disseminate together, with the mesenchymal cells being required for dissemination but not viable at the metastatic site. Thus, the same EMT-dependent metastasis program could underlie the formation of both types of metastasis observed.
3. Figure S1 shows that a higher percentage of cancer cells stain positive for Vimentin or Zeb1 than for α SMA or Fsp1. Using the KPF; R26Dual; α SMA-Cre and Fsp1-Cre models: What is the percentage of Zeb1+ or Vimentin+, EGFP+ cells, i.e. cells undergoing an EMT that would be missed based on α SMA or Fsp1 as a marker?
4. Figure 3A and B: an overlay of α SMA and tdTomato as well as α SMA and EGFP should be shown. The signal intensity of α SMA should be slightly increased or another pseudo-color (white?) could be chosen. It looks as if a fraction of EGFP+ tumor cells in 3B could be α SMA+, which suggests inefficient recombination by α SMA driven Cre.
5. Figure 4: Starting from fully epithelial cells derived from KPF; R26Dual; α SMA-Cre (and also Fsp1-Cre) mice, how long does it take until α SMA and Fsp1 are significantly up regulated in respect to other mesenchymal markers? How long does it take until the majority of cells are tdTomato+? Are α SMA and Fsp1 suitable markers to capture early, i.e. partial EMT-stages?
6. Figure 2B and 6C: The authors observed an interesting population of tdTomato, EGFP double positive cells. Representative images should be shown. Taking all EGFP+ cells together, approximately 20-25% of these cells are double positive, meaning a good proportion of cancer cells have initiated the EMT-program just shortly before sacrifice. Could EMT be a late event in the PDAC model? Have these cells still retained E-cadherin expression? Which mesenchymal markers (e.g. Zeb1, Vimentin) do these cells express apart from α SMA or Fsp1?
7. Page 7, 1st paragraph, last sentence: GFP+/dTom+ cells indicate a cell that is just undergoing an EMT and GFP as protein remains due to its protein stability. These cells do not represent an early EMT, as stated by the authors. Or is SMA an early EMT marker?
8. Figure 3C: The GFP channel should be shown individually.
9. Figure 4C: The in vitro EMT seems to work quite efficiently at the cell morphological level, yet seem markers are not changing (e.g. Vim). Any conclusions why EMT is highly efficient in vitro, yet rare in vivo? Or is the lineage tracing system failing in vivo?
10. Figure 5 is not really adding any new insights. Here, all cells of the mouse are red, and the resolution and interpretation of the experiments is even more obscured.

11. Figure S6: an overlay of α SMA and tdTomato should be shown. The recombination efficiency of Fsp1-Cre in stromal fibroblasts could be low. An immunofluorescence staining of Fsp1 on KPF; R26Dual; Fsp1-Cre mice should be performed to confirm efficient recombination of Fsp1 driven Cre. To confirm the minimal overlap of α SMA and Fsp1 in stromal fibroblasts, stainings for Fsp1 and α SMA on the same or on serial sections should be performed.

12. Overall, the mouse numbers used for the study (three mice per model) is relatively low and might not capture the whole spectrum of well to poorly differentiated PDAC.

13. The amount of imaging fields analyzed (three on average) is low, especially since positive events are rare.

Minor comments:

1. In the Abstract, the authors state that: "Direct functional studies and precise determination of the frequency of partial EMT program in tumors needs unraveling." What does the term "Direct functional studies" refer to? The presented study is mostly descriptive.

2. In the Methods section, the authors should state the origin of the R26Dual mice.

3. Page 8, row 4: It says: "Further, the specificity and functional relevance of the dual-recombinase fluorescent lineage tracing was examined by immunolabeling tumor sections..." What does "functional relevance" refer to?

4. Figure 2D: Has the CK19 channel been included in the Merge?

5. The formatting of the scale bars is inconsistent throughout multiple figures.

6. Figure 3A and 3B: A thin scale bar can still be seen below the thick bar in the merge of 3A. Likewise, the panels of individual channels in 3A and 3B show very faint scale bars.

7. Figure 5B: In the R26mT/mG system all cells should be either EGFP or tdTomato positive. However, there are cells with no clearly visible color.

8. Figure S1: Scale bars are missing. The figure legend should state how many imaging fields were analyzed per mouse.

9. Figure S4C: The figure legend states the scale bar represents 50m, while the label in the actual panel says 20m.

10. Figure S6: It should say Pdx1-Flp, not Cre.

1st Revision - authors' response

01 June 2018

Response to the reviewers:

Referee #1 (Remarks for Author):

The study of Chen et al. describes results of dual reporter lineage tracing of metastasis forming cells in pancreatic cancer (PDAC). Previously the same group has provided results that challenged the concept of EMT-mediated metastasis by showing that PDAC metastases formation does not depend on the EMT transcription factors Snail or Twist. Here, to better trace the transient events of EMT potentially contributing to metastasis, the authors used a genetically engineered mouse models, where the epithelial cancer cells irreversibly lose EGFP expression and gain tdTomato expression upon the expression of the EMT markers α SMA or Fsp1. The authors conclude that the metastatic lesions exhibit either an exclusive epithelial or partial EMT phenotype, whereas the so-called partial EMT-phenotype is limited to individual cells and micrometastasis, but excluded from macrometastatic lesions. To confirm this still provocative result, they use another mouse model, where only the cells that induce α SMA lose tdTomato and gain EGFP. The study is interesting,

clearly presented and provides further evidence for the dispensability of EMT for the metastasis formation.

Specific comments:

1. Figure 1 nicely shows the PDAC progression in the used model by immunohistochemistry (ihc). An informative addition would be ihc characterization of both the primary tumors and metastatic tissues with CK19 as well as aSMA & FSP1.

- Excellent suggestion. As suggested by this reviewer, we included in Figure 1 CK19 IHC for both primary tumor (Figure 1B) and liver metastasis (Figure 1C). The aSMA and FSP1 staining of primary tumors is shown in Figures 3A-C, 6E, Supplementary Figures S1A, S2 and S6. We also include IHC for aSMA and FSP1 in primary tumors and liver metastases (**NEW Supplementary Figure S3**). These results corroborate the findings that aSMA and FSP1 are detected in the cancer cells associated with the primary tumors but are not noted in macrometastases.

2. Figure 2A and 6C show even stronger CK19 in the aSMA+ and FSP1+ cells with gained tdTomato, compared to the neighboring epithelial cancer cells, whereas in Figure 5B CK19 is faint in the FSP1+ cancer cell with gained EGFP in the area of otherwise strongly CK19+ epithelial cancer cells: How specific is CK19 detection?

- This is a valid point. We showed in Figures 2A-B and 6B-C that CK19 immunolabeling captures 95% of lineage traced (EGFP⁺) cancer cells in primary tumor, as well as faithfully detects lineage traced (EGFP⁺) cancer cells in metastasis (Figures 1C, 2D, 5B-C, 6D). Notably, the CK19 signal in our study was evaluated via the 647nm channel, whereas the tdTomato signal was evaluated via the 581nm channel, preventing any overlap in fluorescent signal and ensuring accuracy and specificity of the CK19 immunofluorescent detection. Should CK19⁺ cells be detected without EGFP expression, these cells were tdTomato⁺, indicative of mesenchymal transition (as shown in Supplementary Figure S4A-B). The expression of CK19 is heterogeneous, and we have emphasized this point in the revised manuscript (**page 7**: “To confirm these findings while keeping in mind the heterogeneous levels of expression of CK19, we carried out similar immunolabeling experiments using E-cadherin to capture epithelial cancer cells. Similar results were obtained, wherein E-cadherin labeling captured the majority of EGFP expressing cancer cells (**NEW Supplementary Figure S4C**).”). Finally, the specificity of the CK19 labeling is also evidenced by the lack of signal in the secondary antibody control panel (**Reviewer Figure 1B**). Thank you for bringing this to our attention.

3. Is CK19 expression retained upon full EMT in the PDAC cells? The authors should show CK19 mRNA and protein expression changes upon TGF- β -induced EMT in Figure 4 B and C. Do the vimentin+/Zeb1+/FSP1+ and YFP+ cells express CK19 in Figure S1A-C? Since the detection of cancer cells rely on CK19, this should be clarified.

- As suggested, we have now included qPCR analyses and immunolabeling for CK19 transcript/protein levels in cancer cells subjected to TGF- β treatment (**NEW Figure 4C and Supplementary Figure S5B**). The results show that CK19 expression is reduced in cancer cells subjected to TGF- β treatment, yet detectable at low levels. With respect to the data shown in Supplementary Figure S1A-C, we point the reviewer to the lineage tracing strategy used in these experiments. The YFP expression captures all cancer cells and enables monitoring of EMT program in cancer cells that may present with reduced expression of epithelial markers.

4. The data shows that the macroscopic metastatic lesions have been formed primarily of cells that have not experienced EMT, but after undergoing EMT could some mesenchymal cancer cells contribute to the fibroblast population? It should be discussed, what happens to the cells in the mesenchymal cancer cell-positive microlesions; don't they proliferate at all?

- Cancer cells that have experienced EMT in micrometastases are captured using CK19 expression and it appears they do not directly contribute to the fibroblast population (CK19-negative). Our genetic strategy and approach cannot determine whether the cancer cells that have homed to the metastatic site and present with an EMT program (micrometastases), display fibroblasts-like functions. This interesting point was discussed in the revised manuscript (**page 13**: “Although our current study did not functionally characterize the cancer cells that present with a partial EMT

program, it is possible that such cells may exert paracrine, fibroblast-like functions and contribute to formation of large metastatic nodules.”).

In our studies, we detect and capture endogenous fluorescent signal (EGFP and tdTomato) in addition to CK19 immunofluorescence and DAPI labeling of nuclei. It is technically impossible to embed in this analysis another marker to denote the cells' proliferation status. Accordingly, we revised our discussion (**page 13**): “The larger size of non-EMT metastases suggests a potentially proliferative feature of such metastatic nodules. This would certainly align with the notion that a proliferative program is likely suspended in the cancer cells with an EMT program (Ozdemir et al. 2014; Zheng et al. 2015). Further, a recent study highlighted that epithelial like breast cancer cells expressing Epcam were more proliferative than those that did not express Epcam and classified as subpopulations of cancer cells with EMT program (Pastushenko et al. 2018). Our study also suggests that the cancer cells with a partial EMT program, as noted in the micrometastases, may reflect latent, and perhaps more therapeutically recalcitrant metastatic cancer cells (Shah et al. 2007; Yin et al. 2007; Arumugam et al. 2009; Wang et al. 2009; Hugo et al. 2016). It is intriguing to propose that such rare, perhaps more latent metastatic cancer cells that arose via acquisition of a partial EMT program reflect cancer stem cell-like phenotype associated with pancreatic cancer (Valle et al. 2018); although this proposition needs experimental evidence.”

5. On page 9, the authors refer to Figure S4C to indicate the existence of EGFP+ α SMA-expressing metastatic cancer cells as single cells or micrometastasis, but this figure only shows EGFP+ myofibroblasts and EGFP- cancer cells.

- We apologize for this error and thank you for pointing it out to us. We have revised the manuscript to correctly represent the data show in Figure S4C (now as revised **New Supplementary Figure S6C**), **page 9**: “Established macrometastases in KPF; α SMA-Cre;R26^{mT/mG} mice exclusively revealed an epithelial phenotype, with ubiquitous tdTomato expression and without any evidence for EGFP expression (Figure 5C, Table 1). Non-EMT CK19⁺ cancer cells could also be observed as single cells in the lung of KPF; α SMA-Cre;R26^{Dual} mice (Table 1, Supplementary Figure S6C). EGFP-expressing metastatic cancer cells were observed only as single cancer cells or part of micrometastases (colonies of 3-5 cancer cells, Figure 5C and Table 1).”.

6. Figure S3B lacks the red signal mark in the single channel image (tdTomato).

- Corrected (now Supplementary Figure S4B), thank you for pointing out also.

Referee #2 (Remarks for Author):

In their Manuscript, Chen et al provide evidence for an EMT independent metastatic process in pancreatic cancer. By employing a Dual-Recombinase System and mesenchymal cell reporters (α SMA-Cre and Fsp1-Cre) within genetically engineered mice, they show that partial EMT program is restricted to isolated single cancer cells building micrometastases (3-5 cancer cells), while established macrometastases preserve a purely epithelial phenotype, indicating two distinct and mutually exclusive types of metastatic dissemination programs.

This is a well-designed study and the authors are to be commended for their deep investigation. The manuscript is also well written and the topic is certainly of interest.

Considering extensive focus on the concept of EMT, I would expect some more in-depth discussion on the properties of cells with partial EMT, apart from their ability to form micrometastases, for example stem cell-like properties and role in chemoresistance.

- An excellent point, we have significantly revised our discussion to expand upon the potential functions/roles of partial EMT program (**page 12-13**).

Also, some reference in studies of human PDAC material reporting a similar phenomenon (in form of tumor budding) would help to further understand the contribution of partial EMT program in cancer cell dissemination.

- We have now included additional references and discussion points with respect to human PDAC literature and partial EMT program in systemic dissemination (**page 12-13**).

In a recently published study (Krebs et al, Nat Cell Biol. 2017 May;19(5):518-529), another group has shown that in the same pancreatic cancer model, driven by Pdx1-cre-mediated activation of mutant Kras and p53 (KPC model), the EMT-TF Zeb1 is a key factor for the formation of precursor lesions, invasion and metastasis. The authors could perhaps more extensively comment on the possibility of different EMT transcription factors being implicated in driving metastasis.

- We now include a section of our discussion regarding the study published by Krebs et al. (**page 12-13**), thank you for this insightful comment. The complete genetic deletion of Zeb1 in PDAC GEMM was found to be dispensable for the formation of metastases, similar to the findings we reported using Snail or Twist conditional knockouts (Zheng et al. Nature, 2016), albeit 50% reduction in metastasis is observed when Zeb1 is absent. In this study, 50% of KPC mice exhibited metastasis despite complete loss of Zeb1. Additionally, bulk of the experiments were not performed using spontaneous genetically engineered mouse models but transplanted cell lines. The heterogeneity of EMT in cancer progression was also highlighted in a recent study by Pastuschenko et al. (Nature, 2018). We have revised our discussion (**page 12-13**) to highlight these important points.

Referee #3 (Comments on Novelty/Model System for Author):

The lineage tracing models used by the authors do not adequately address the problem, they rather reproduce the uncertainties and problems of previously published lineage tracing experiments. The results thus are not adding novel insights to the current debate on EMT and metastasis.

- We respectfully disagree with this statement. Our models are the state of art and provide testing of EMT requirement for metastasis. At the very center of the EMT debate (and historical progression in our scientific understanding of the biology of EMT in cancer), is the critical evaluation of the monitoring of partial EMT program in cancer cells in the tumors and in the secondary metastatic tumors. The immunolabeling of antigens likely capture the gain of mesenchymal markers/cellular features and the loss of epithelial markers (Brabletz, Kalluri, Nieto & Weinberg, Nature Reviews Cancer, 2018; Mittal, Annual Review of Pathology, 2018, Kumar Jolly et al. Molecular Oncology, 2017). But this immunolabeling often leads to potential under-representation of cells undergoing partial EMT since it relies on the capture of cells presenting both epithelial and mesenchymal markers. Further, this strategy does not enable the capture of the history of EMT program at the metastatic lesions because most cancer cells are proposed to have undergone MET program, thereby annihilating any evidence that the EMT-MET program actually occurred. With lineage tracing, a more accurate quantitation of EMT program in cancer cells can be realized (Rhim et al., Cancer Cell, 2012). Nevertheless, the model used in Rhim et al., (Cancer Cell, 2012) could not lineage trace both epithelial and mesenchymal phenotypes. The strategy presented here is an approach in which acquired EMT program in tumor is documented without losing such evidence despite MET at the metastatic site. This represents a significant advance in our evaluation of partial EMT program in metastasis.

Referee #3 (Remarks for Author):

The manuscript submitted by Chen et al. reports results that raise the possibility of an EMT-dependent micrometastasis program and a EMT-independent macrometastasis program in the well-established KP metastatic mouse model of PDAC. The authors used a genetic dual recombinase-based and a more conventional Cre-reporter-based lineage tracing approach to genetically mark cancer cells that have undergone an EMT. Here, they also demonstrate that the KP mouse model can be driven by PDX-Flippase and Frt-flanked alleles of activated Ras and p53 to induce carcinogenesis. EMT marker driven Cre expression is then used to induce a color switch in cancer cells undergoing an EMT from GFP to dTomato. However, in this system, mesenchymal cells of the tumor stroma will also gain expression of dTomato. Cancer cells with an EMT program could be found in the primary tumor and as single disseminated cells or micrometastasis in lung and liver. In contrast, established macrometastasis consisted of cells that never had activated an EMT. Based on these findings, the authors conclude that two distinct and mutually exclusive types of metastatic

dissemination programs have been functional and they suggest the existence of a novel EMT-independent metastasis program.

The authors' observations are certainly interesting. However, the results should be seen in the context of the current debate on the use of certain EMT markers and Cre-driver lines to actually detect and visualize cancer cells undergoing an EMT *in vivo*. Here, the manuscript does not provide additional new insights.

- We believe that our models use the same Cre driver and EMT marker that others have used in their models (Rhim et al., *Cancer Cell*, 2012; Aiello et al., *Nature Communications*, 2016). Moreover, we used multiple methods to track EMT program. Respectfully, we are surprised by the summary dismissal of our experimental system, which does not allow for constructive and informed discussion about the functional role of EMT in metastasis.

First of all, the authors use SMA-Cre driver line to visualize cancer cells that have undergone an EMT. A recent comment by Aiello et al. (2017) in *Nature* shows that SMA is "an unreliable marker of EMT in the KPCY mouse model" and they question the suitability of this line for the adequate detection of an EMT. The authors here have used in addition a Fsp1-Cre driver line to visualize cancer cells that have undergone an EMT. While Aiello et al. demonstrate that Fsp1 is expressed in >20% of the cells in a KPC tumor, here only 2.7% of cells are detected. Based on expression of Fsp1, Zeb1 and E-cadherin, Rhim et al. (*Cell*, 2012) report that approximately 42% of cancer cells in the KPC mouse models have undergone an EMT, meaning EMT would not be a rare event.

- The reviewer cites a commentary, non-traditionally peer reviewed opinion (brief communication arising, BCA) letter (Aiello et al, *Nature BCA letter*, 2017) to discount the value of aSMA as valid marker of EMT in tumors. We would like to point out to the reviewer that we did not offer a response to the BCA opinion letter because we chose to publish our findings in a traditional peer-review process, such as this manuscript. We point this reviewer to the experiments we carried out using the identical anti-aSMA antibodies that were used in Aiello et al. (*Nature Communications*, 2016) and report the findings in addition to data collected using distinct anti-aSMA antibodies (Supplementary Figure S2). Of note, the data shown in the opinion BCA letter (Aiello et al, 2017) immunolabeled tumors for YFP rather than capturing endogenous YFP expression, and the authors showed highly heterogeneous labeling of tumor tissue. These data do not offer sufficient evidence to unequivocally dismiss aSMA expression as a mesenchymal marker gained by cancer cells undergoing EMT program in the context of pancreatic cancer.

We refer the reviewer to Aiello et al. (*Nature Communications*, 2016), in which FSP1⁺ EMT cancer cells were noted at a frequency of ~10%, albeit with a range from 0 to 20%. Notably, this result by Aiello et al. (*Nature Communications*, 2016) is discordant with Aiello et al. (2017) opinion BCA letter in which they report a frequency of FSP1⁺ EMT cancer cells at ~20%, with a range from 0 to 70%. The reviewer has misunderstood our findings: we showed a range of 2% to 7.5% (mean: 4%) of FSP1⁺ YFP⁺ (lineage traced cancer cells, Supplementary Figure S1A). The 2.7% reported in Figure 6C reflects the percent of CK19⁺ FSP1⁺ cancer cells out of all CK19⁺ cancer cells in the field of view.

Finally, we would like to point the reviewer to the details of the quantifications made in Rhim et al. (*Cell*, 2012). In this study, the authors clearly listed that: "We then used the YFP lineage label to identify PDAC cells that had completed an EMT. Since labeling was limited to cells of epithelial origin, we defined EMT as having occurred if a cell co-expressed YFP and either Zeb1 (Fig. 1P) or Fsp-1 (Supp Fig. 1D) and/or lacked E-cad (Fig. 1Q) expression. Using this approach, we observed that 42% of the lineage labeled YFP⁺ cells in PKCY tumors had undergone EMT (Fig. 1P)" This quantification (42%) is thus defining EMT cancer cells using antibody-based immunolabeling rather than a strict definition criterion we applied to our analysis to remove subjectivity.

In addition, the expression of SMA and FSP1 do not substantially overlap questioning the mesenchymal cell-specific expression of these markers. Along these lines, in the Introduction the authors cite the reports by Aiello et al. (2016 and 2017) that Fsp1-positive cells are found in macrometastases in the KPC model. However, it is not addressed later on why the lineage tracing does not recapitulate this observation.

- The lack of overlap between α SMA and FSP1 expression is noted in fibroblasts (Supplementary Figure S8A), whereas we report an overlap in α SMA, Vimentin and Zeb1 in FSP1-Cre lineage traced cancer cells (Supplementary Figure S7B-D). Nonetheless, we now include immunolabeling of tumors from the FSP1-Cre lineage tracing experiments for FSP1 (**NEW Supplementary Figure S8C**). We also immunolabeled tumors for α SMA and FSP1 (**NEW Supplementary Figure S8B**). These results showcase that the FSP1-Cre lineage tracing strategy we employed faithfully capture the FSP1 immunolabeled cell population, and that the fibroblasts do not present with α SMA and FSP1 co-localization. We removed the reference to Aiello et al. 2016 as this article does not address macrometastases.

In the BCA opinion letter by Aiello et al. (2017), the authors immunolabeled FSP1 in macrometastases using an antibody. Our approach, using FSP1-Cre lineage tracing, did not rely on immunolabeling to capture FSP1⁺ cells. We speculate that the FSP1⁺ cancer cells observed in the macrometastases in Aiello et al. (BCA opinion letter, 2017) could reflect inaccurate labeling by the antibodies used, which aligns with the extreme variation these authors observed across their animals (note the error bar for the % FSP1⁺ cancer cells in macrometastases in Aiello et al. 2017, Figure 2e, which exceeds that of the mean, supporting that multiple of the macrometastases they evaluated had no FSP1⁺ cancer cells).

A major drawback of the lineage tracing experiments used here is that not only cancer cells can gain tdTomato expression upon Cre-mediated recombination but also all mesenchymal cells in the tumor. This fact makes the analysis rather ambiguous, as can also be seen by the rather poor quality of marker co-staining shown throughout the manuscript. Higher resolution and three-dimensional reconstructions are required to obtain data which convincingly support the authors' conclusions.

- It is unclear how the lineage tracing experiments is stated to bring about ambiguity in our analyses. We employed CK19 to confirm the detection of cancer cells and excluded myofibroblasts or other fibroblasts. We provided high resolution images for all of our analyses. We attempted to carry out 3D-reconstructions, however the endogenous fluorescent labels are quenched in the 3D multi-layer scanning process, precluding such analyses.

Finally, the lineage tracing experiments are somehow leading themselves "ad absurdum" by using the epithelial marker CK19 to detect cancer cells. The authors escape this problem in stating that dTomato+/CK19+ cells are in a "partial EMT". However, no further characterization is offered to support this notion, also to characterize the single EMT cancer cells in micrometastasis in lung and liver. In the past, the use of cytokeratins has been a major roadblock in studying EMT in cancer progression, because it may completely ignore any EMT cells.

Since the levels of EMT cancer cells detected by dTomato and by CK19-expression are rather low at ~2 cells per field in a highly aggressive primary tumor, one wonders whether these cells are really EMT cells. They should be further characterized to prove their identity. Is CK19 expression retained during the EMT processes marked by α SMA or Fsp1? Are CK19 positive cells with an EMT program also E-cadherin positive? The authors should comment on this.

- We respectfully disagree and point to the reviewer that our lineage tracing for mesenchymal marker and CK19⁺ cancer cells enabled evaluation of all metastases, as validated by H&E staining (Figure 2C-D), found in the lungs and liver of our GEMMs. It is unclear what studies have challenged the use of cytokeratin 19 as a marker for pancreatic cancer cells as this marker is widely accepted to robustly capture pancreatic cancer cells (Zapata M. et al. (2007) Cytojournal; Wagner M. et al. (2001) Genes and Development; Jain R et al. (2010) Appl Immunohistochem Mol Morphol) and disseminated cancer cells in PDAC GEMMs were noted to express CK19 (Rhim et al (2012) Cell, Figure 4b). We reported consistent detection of CK19 in cancer cells, including cancer cells with a mesenchymal phenotype (Figure 2A, D, Figure 5B-C, Figure 6B-D). Critically, in macrometastatic nodules employing CK19 expression, we failed to capture any cells that would have undergone EMT-MET. Nevertheless, we now include E-cadherin immunolabeling analyses, which confirm the findings employing CK19 immunolabeling (**NEW Supplementary Figure S4C**).

Specific comments:

1. Importantly, EMT-independent mechanisms of tumor dissemination have been well described. Hence, the report of a possible EMT-independent metastasis program is not novel.

- We respectfully disagree. Circulating tumor cell analyses reporting on the detection of cancer cells with hybrid E-M phenotypes or epithelial-like phenotype in the blood during metastatic dissemination do not necessarily implicate them as the cancer cells that will ultimately form secondary/metastatic tumors lacking EMT program. Our study is innovative and used new informative models to address the role of EMT in metastasis.

2. The authors observed cells that have undergone an EMT as single disseminated cells or in micrometastatic lesions exclusively. Were these lesions exclusively single or also multi-colored? It is possible that epithelial and mesenchymal cancer cells disseminate together, with the mesenchymal cells being required for dissemination but not viable at the metastatic site. Thus, the same EMT-dependent metastasis program could underlie the formation of both types of metastasis observed.

- The micrometastatic lesions or single disseminated cells were noted to arise via EMT or non-EMT program (please see Table 1). We did not observe a macrometastatic lesion that exhibited an EMT program and these macrometastatic lesions were strictly non-EMT derived (Table 1). Within a given micrometastatic lesion, cancer cells were strictly either of the EMT or non-EMT phenotype, we clarified this point. Our data does not allow for the speculation that EMT-derived cancer cells are functionally relevant for the metastatic potential of the non-EMT cancer cells. A recent study by Patuscheke et al. (Nature, 2018) suggest that cancer cells that maintain an epithelial phenotype are more proliferative than those with a mesenchymal phenotype. This intriguing finding is discussed in the context of our observations (page 13).

3. Figure S1 shows that a higher percentage of cancer cells stain positive for Vimentin or Zeb1 than for α SMA or Fsp1. Using the KPF; R26Dual; α SMA-Cre and Fsp1-Cre models: What is the percentage of Zeb1+ or Vimentin+, EGFP+ cells, i.e. cells undergoing an EMT that would be missed based on α SMA or Fsp1 as a marker?

- Supplementary Figure S1 does not depict the percentage of cancer cells positive for α SMA. We point the reviewer to the data in Supplementary Figure S7B-D, in which we list a high percentage of overlap of FSP-Cre-lineage traced cancer cells for α SMA (>60%), vimentin (>50%), and Zeb1 (30%).

4. Figure 3A and B: an overlay of α SMA and tdTomato as well as α SMA and EGFP should be shown. The signal intensity of α SMA should be slightly increased or another pseudo-color (white?) could be chosen. It looks as if a fraction of EGFP+ tumor cells in 3B could be α SMA+, which suggests inefficient recombination by α SMA driven Cre.

- This is an excellent point. In Figure 3A-B, we included all individual panels to clearly demonstrate the co-localization of α SMA and tdTomato. As requested, we now include the specific α SMA and tdTomato overlay (NEW Figure 3A-B). As requested, we use white for the pseudo-coloring of α SMA, which more clearly demonstrates efficient recombination driven by α SMA-Cre.

5. Figure 4: Starting from fully epithelial cells derived from KPF; R26Dual; α SMA-Cre (and also Fsp1-Cre) mice, how long does it take until α SMA and Fsp1 are significantly up regulated in respect to other mesenchymal markers? How long does it take until the majority of cells are tdTomato+? Are α SMA and Fsp1 suitable markers to capture early, i.e. partial EMT-stages?

-We listed 48 and 96hrs time points for TGF- β treatment (see Methods). Our experiments indicate a significant upregulation of α SMA (*Acta2*) and FSP1 (*S100A4*) transcripts levels in cells following TGF- β treatment, along with *Fnl*, *Colla1*, *Twist* and *Snail* (Figure 4C).

6. Figure 2B and 6C: The authors observed an interesting population of tdTomato, EGFP double positive cells. Representative images should be shown. Taking all EGFP+ cells together, approximately 20-25% of these cells are double positive, meaning a good proportion of cancer cells have initiated the EMT-program just shortly before sacrifice. Could EMT be a late event in the PDAC model? Have these cells still retained E-cadherin expression? Which mesenchymal markers (e.g. Zeb1, Vimentin) do these cells express apart from α SMA or Fsp1?

- In Figures 2B and 6C, we report the detection of cells which are triple positive (EGFP+ tdTomato+ CD19+). Such a cell is represented in Figure 2A, and 6B-C. We show that out of all CK19+ cells

(with nearly all are also EGFP⁺), ~1.8% of these are tdTomato⁺ (mesenchymal), and ~0.5% are triple positive (EGFP⁺ tdTomato⁺ CD19⁺). It is unclear how the reviewer computed 20-25% double positive cells from these graphs (Figures 2B and 6C). We point the reviewer to the data in Supplementary Figure S7B-D, in which we list a high percentage of overlap of FSP-Cre-lineage traced cancer cells for αSMA (>60%), vimentin (>50%), and Zeb1 (30%). E-cadherin immunolabeling analyses were also carried out, confirming the findings obtained using CK19 (**NEW Supplementary Figure S4C**).

7. Page 7, 1st paragraph, last sentence: GFP+/dTom+ cells indicate a cell that is just undergoing an EMT and GFP as protein remains due to its protein stability. These cells do not represent an early EMT, as stated by the authors. Or is SMA an early EMT marker?

- The remaining EGFP protein detection despite onset of αSMA-Cre mediated recombination and expression of tdTomato is indicative of an αSMA-driven EMT program in its early stage. Whether αSMA could be considered an early marker of EMT cannot be conclusively drawn from our results since other mesenchymal markers could be expressed in these cells. The data can only support that these cells recently expressed epithelial driven expression marker, EGFP. We have revised our results section to clarify this point: “Furthermore, a discrete proportion (0.5%) of EGFP/tdTomato double-positive cells (with diminishing EGFP and emerging tdTomato expression) were also observed (Figure 2B), possibly reflecting retained EGFP proteins despite start of tdTomato transcription in cancer cells at the onset of the αSMA-Cre driven partial EMT phenotype” (**page 7**).

8. Figure 3C: The GFP channel should be shown individually.

- Done.

9. Figure 4C: The *in vitro* EMT seems to work quite efficiently at the cell morphological level, yet seem markers are not changing (e.g. Vim). Any conclusions why EMT is highly efficient *in vitro*, yet rare *in vivo*? Or is the lineage tracing system failing *in vivo*?

- We can only conclude that *in vitro* EMT induction by TGFβ, which leads to robust induction of many mesenchymal markers as shown in Figure 4 (between 3 and 10-fold increase in FSP1 (*S100A4*), αSMA (*Acta2*), *Fnl*, *Colla1*, and *Snail* transcript levels), differs from what is observed *in vivo*. This is not a novel concept, and many EMT researchers can only cautiously interpret findings observed *in vitro* since EMT *in vivo* is far more complex and nuanced. We monitored cancer cells with EMT program in the primary tumor (Figures 2-6, Supplementary Figures S2-7, Table 1) and via vimentin immunolabeling (Supplementary Figure S7B, D).

10. Figure 5 is not really adding any new insights. Here, all cells of the mouse are red, and the resolution and interpretation of the experiments is even more obscured.

- The fact that tdTomato to EGFP switch was used in addition to the EGFP to tdTomato switch to monitor EMT is a major strength of our study, as it avoids bias of distinct fluorescent protein detection in tissue. In addition, the R26^{mT/mG} reporter allele has been widely used by numerous studies that support the efficacy of the cre-recombinase mediated tdTomato to EGFP fluorescence switch. The addition of this well-documented reporter allele serves as a further confirmation of our observations in addition to the herein newly established R26^{Dual} reporter.

11. Figure S6: an overlay of αSMA and tdTomato should be shown. The recombination efficiency of Fsp1-Cre in stromal fibroblasts could be low. An immunofluorescence staining of Fsp1 on KPF; R26Dual; Fsp1-Cre mice should be performed to confirm efficient recombination of Fsp1 driven Cre. To confirm the minimal overlap of αSMA and Fsp1 in stromal fibroblasts, stainings for Fsp1 and αSMA on the same or on serial sections should be performed.

- Excellent point. The requested overlay is now shown in the **NEW Supplementary Figure S8A**. The requested immunolabeling for FSP1 in the FSP1-Cre mice was carried out, as well as αSMA and FSP1 double immunolabeling. These results are shown in the **NEW Supplementary Figure S8B-C** and they confirm robust efficacy of the FSP1-Cre transgene to capture FSP1 expressing cells and also confirm the lack of overlap in αSMA and FSP1 expression in fibroblasts but overlap in EMT cancer cells (as was shown in Figure 6E and Supplementary Figure S7D).

12. Overall, the mouse numbers used for the study (three mice per model) is relatively low and might not capture the whole spectrum of well to poorly differentiated PDAC.

- We appreciate the comment. The generation of these complex GEMMs prevents us from increasing the mouse numbers per group. Nevertheless, the fact that we used distinct reporter to arrive at the same conclusion should alleviate the issue regarding the number of mice used (Table 1).

13. The amount of imaging fields analyzed (three on average) is low, especially since positive events are rare.

- We appreciate the comment: we screened lungs and liver embedded tissues for metastases by serial sectioning of the entire tissue. All mice presented with metastases and our approach remains unbiased and enables robust quantification of the reported events.

Minor comments:

1. In the Abstract, the authors state that: "Direct functional studies and precise determination of the frequency of partial EMT program in tumors needs unraveling." What does the term "Direct functional studies" refer to? The presented study is mostly descriptive.

- Corrected to: "Precise determination of the frequency of partial EMT program in tumors and metastases needs unraveling."

2. In the Methods section, the authors should state the origin of the R26Dual mice.

- Corrected.

3. Page 8, row 4: It says: "Further, the specificity and functional relevance of the dual-recombinase fluorescent lineage tracing was examined by immunolabeling tumor sections..." What does "functional relevance" refer to?

- Corrected to: "Further, the specificity of the dual-recombinase fluorescence lineage tracing system was examined by immunolabeling tumor sections of KPF; α SMA-Cre;R26^{Dual} mice using antibodies to α SMA"

4. Figure 2D: Has the CK19 channel been included in the Merge?

- No, it was not included. The label was changed to reflect this and we apologize for the error.

5. The formatting of the scale bars is inconsistent throughout multiple figures.

- Corrected.

6. Figure 3A and 3B: A thin scale bar can still be seen below the thick bar in the merge of 3A. Likewise, the panels of individual channels in 3A and 3B show very faint scale bars.

- Corrected.

7. Figure 5B: In the R26mT/mG system all cells should be either EGFP or tdTomato positive. However, there are cells with no clearly visible color.

- We now include additional images that showcase that the tdTomato expression in cells that appeared to be negative in Figure 5B, due to low expression and technical issues, were indeed positive for tdTomato (**Reviewer Figure 1A-B**). In the presentation of the data in Figure 5B, the cells that exhibited relatively lower tdTomato signal were displayed as negative for tdTomato when the imaging software automatically reduces the brightness of this channel to avoid overexposure/hotspot from cells with higher tdTomato signal within the same visual field. To showcase this point, this feature (automatic reduction in brightness by the imaging software) was turned off and the resulting images (exact same field of view as shown in Figure 5B) are now presented in **Reviewer Figure 1A**. We also include the detailed microscopic parameters for image

capture and image display to evidence this point (graph in **Reviewer Figure 1A**). We revised the methods and accompanying source data file to include the detailed microscopic parameters for image capture and image display for all images in our manuscript (see accompanying source data). Further, we include in **Reviewer Figure 1B** additional controls. No tdTomato signal is noted in tumors from mice without the genetic tracer (KPF littermate control, **Reviewer Figure 1B**) when images are displayed with the automatic reduction in brightness by the imaging software is turned off. The specificity of the CK19 labeling is evidenced by the lack of signal in the Secondary antibody alone panel (**Reviewer Figure 1B**).

8. Figure S1: Scale bars are missing. The figure legend should state how many imaging fields were analyzed per mouse.

- Corrected.

9. Figure S4C: The figure legend states the scale bar represents 50m, while the label in the actual panel says 20m.

- Corrected.

10. Figure S6: It should say Pdx1-Flp, not Cre.

- Corrected.

Reviewer Figure 1. Examination of EMT in KPF; α SMA-Cre;R26^{mT/mG} and KPF control mice.

(A) Original immunofluorescence images (of the image also displayed in **Figure 5B**) of primary PDAC tumors from KPF; α SMA-Cre;R26^{mT/mG} mice examined for intrinsic tdTomato and EGFP signals in combination with CK19 immunofluorescence co-staining. The table below lists the microscopy parameters used for image capture and display. (B) Immunofluorescence images of primary PDAC tumors from KPF mice (negative control for tdTomato and EGFP) examined for intrinsic tdTomato and EGFP signals as well as CK19 staining (compared to secondary-antibody-only control) using the same imaging parameters as in (A). Scale bars in all panels, 20 μ m.

Reviewer Figure 1

2nd Editorial Decision

06 July 2018

Thank you for the submission of your revised manuscript to EMBO Molecular Medicine. We have now received the enclosed reports from the referees that were asked to re-assess it. As you will see the reviewers are now globally supportive and I am pleased to inform you that we will be able to accept your manuscript once you've addressed both referees' comments in writing.

Please submit your revised manuscript within two weeks. I look forward to seeing a revised form of your manuscript as soon as possible.

I look forward to reading a new revised version of your manuscript as soon as possible.

***** Reviewer's comments *****

Referee #1 (Remarks for Author):

The manuscript has improved and the authors have addressed all of my questions.

Based on the provided evidence, it seems clear that the macrometastatic colonies lack EMT traces, suggesting that the previously prevailing EMT-MET concept does not hold in this cancer model.

Just a minor concern remains, even with the clarifying explanations, regarding how complete the detection of primary tumor and metastatic EMT cancer cells is by the used immunostaining and mesenchymal/cancer cell tracers *in vivo*.

The revised Discussion now provides more critical/careful considerations and interpretation of these results. As discussed also in the manuscript, more research will be needed to understand the fate and specific functions of the EMT/mesenchymal cancer cells in the primary tumor, during metastasis processes and drug treatments. In any case, this is an interesting and important study.

Referee #3 (Remarks for Author):

Based on the conflicting data and the hot debate in the field, the manuscript has drawn a number of deep, yet important and constructive criticisms. Overall, the authors have extensively and in most parts adequately responded to the reviewers' comments. They have added substantial amounts of additional data, have improved data quality and presentation and revised the manuscript accordingly. The authors have also included more discussions and more recent references on the conflicting data in the field in the revised Discussion section, also to meet the principal critique on the genetic lineage tracing design and the analysis and interpretation of the results. It is obvious and should be accepted that the authors cannot change their basic strategies in genetic lineage tracing and analysis at this stage of the manuscript. It is thus even more important that the interpretation of the results and their discussion is presented in a thoughtful and balanced manner. This has been only been achieved in parts.

As mentioned in the comments by reviewers # 2 and 3, the lineage tracing models used do not substantially add concise analysis to previously reported results. The authors respectfully disagree, yet they have not adequately addressed this general critical point. An appropriate discussion on the strengths and weaknesses of the strategy used should be included.

This reviewer is still puzzled by the fact that the authors clearly show a decrease of CK19 expression during a TGFbeta-induced EMT of a cell line derived from the lineage-tracing experiment, while the main parts of the manuscript rely on the detection of cancer cells that have undergone an EMT on the expression of CK19. Basically, the authors distinguish cancer cells that have undergone an EMT (and possibly are CK19-) and CK19- stromal cells by cell morphology. For example, in Figure 2 and 3 and in the new Suppl. Figure S6C, GFP+ cells are distinguished by CK19 expression as cancer cell or myofibroblast. What if a cancer cell has lost CK19 expression during an EMT? Here, the manuscript is still limited in concise analysis and appropriate interpretation.

The new staining for E-cadherin in primary tumors (Suppl. Figure S4C) shows that tdTomato+ cells have lost E-cadherin expression, thus technically validating the actual lineage tracing method, but it does not add any insights on the distinction between CK19- cancer cells and stromal cells.

The authors still categorize the few tdTomato+ cancer cells that are found in micrometastasis as having undergone a partial EMT. They now provide more discussion on the definition of a partial EMT in the Discussion section, yet they do not really show hard data to demonstrate a partial EMT in the "EMT-like" cells found in micrometastases.

The authors now include a discussion of the report by Krebs et al. (2018) which shows reduced metastasis by genetic ablation of Zeb1 expression in the KPC mouse model of PDAC. However, it is stunning to see that they interpret the results by Krebs et al. (2018) as Zeb1 being "dispensable" for metastasis formation. The authors should not ignore the redundancy between the various EMT core

transcription factors. Moreover, the experiments reported by Krebs et al. analyze the functional contributions of Zeb1 to tumor progression and metastasis and not explicitly to EMT.

2nd Revision - authors' response

12 July 2018

Referee #1 (Remarks for Author):

The manuscript has improved and the authors have addressed all of my questions.

- We appreciate your guidance in the previous review cycle.

Based on the provided evidence, it seems clear that the macrometastatic colonies lack EMT traces, suggesting that the previously prevailing EMT-MET concept does not hold in this cancer model.

Just a minor concern remains, even with the clarifying explanations, regarding how complete the detection of primary tumor and metastatic EMT cancer cells is by the used immunostaining and mesenchymal/cancer cell tracers in vivo.

- This is a relevant and a very important question that has vexed the field of EMT. Many studies used immunohistochemistry and a single cancer cell lineage tracing strategy to conclude on the absolute role of EMT in metastasis. We agree that in the setting of negative results, the bar is usually high. Nevertheless, FSP1 immunolabeling was used in PDAC tumors to show that it robustly captures cancer cells with EMT program (Rhim et al., *Cell* 2012). But this issue must be kept in mind when EMT program being studied at the metastatic loci. We have revised our discussion to further highlight the potential limitations of EMT detections strategies when facing a heterogeneous program such as EMT of cancer cells.

The revised Discussion now provides more critical/careful considerations and interpretation of these results. As discussed also in the manuscript, more research will be needed to understand the fate and specific functions of the EMT/mesenchymal cancer cells in the primary tumor, during metastasis processes and drug treatments. In any case, this is an interesting and important study.

- Thank you. Good points and we concur.

Referee #3 (Remarks for Author):

Based on the conflicting data and the hot debate in the field, the manuscript has drawn a number of deep, yet important and constructive criticisms. Overall, the authors have extensively and in most parts adequately responded to the reviewers' comments. They have added substantial amounts of additional data, have improved data quality and presentation and revised the manuscript accordingly. The authors have also included more discussions and more recent references on the conflicting data in the field in the revised Discussion section, also to meet the principal critique on the genetic lineage tracing design and the analysis and interpretation of the results. It is obvious and should be accepted that the authors cannot change their basic strategies in genetic lineage tracing and analysis at this stage of the manuscript. It is thus even more important that the interpretation of the results and their discussion is presented in a thoughtful and balanced manner. This has been only been achieved in parts.

- We have used state of the art technology and multiple lineage tracing methods to arrive at our cautious conclusions. Our interpretation and conclusion are driven by experimental data. We appreciate the thoughtful comments of the reviewer.

As mentioned in the comments by reviewers # 2 and 3, the lineage tracing models used do not substantially add concise analysis to previously reported results. The authors respectfully disagree, yet they have not adequately addressed this general critical point. An appropriate discussion on the strengths and weaknesses of the strategy used should be included.

- We respectfully disagree that our lineage tracing models do not add substantially to previously reported results. We used a novel dual recombinase GEMM system to lineage track EMT using multiple mesenchymal drivers. This approach was not reported before. The reviewer does not provide any evidence to the contrary or indicating that our models are not superior to previously reported models. The discussion includes possible limitations of our approach.

This reviewer is still puzzled by the fact that the authors clearly show a decrease of CK19 expression during a TGFbeta-induced EMT of a cell line derived from the lineage-tracing experiment, while the main parts of the manuscript rely on the detection of cancer cells that have undergone an EMT on the expression of CK19. Basically, the authors distinguish cancer cells that have undergone an EMT (and possibly are CK19-) and CK19- stromal cells by cell morphology. For example, in Figure 2 and 3 and in the new Suppl. Figure S6C, GFP+ cells are distinguished by CK19 expression as cancer cell or myofibroblast. What if a cancer cell has lost CK19 expression during an EMT? Here, the manuscript is still limited in concise analysis and appropriate interpretation.

- *In vitro* treatment of cancer cells with TGFb to induce EMT resulted in a ~0.3-fold decrease in CK19 transcript levels (Figure 4C), therefore CK19 expression is still detected. The *in vivo* capture of EMT was possible via detection of CK19 in cancer cells, a critical feature of a partial EMT program. The use of CK19 in the study of metastases was critical to ensure their distinction from local myofibroblasts. If the reviewer is postulating the existence of a partial EMT program wherein CK19 would be completely lost, potentially resulting in an underestimation of partial EMT capture in the primary tumor, it would not change the fact that CK19⁺ cancer cells observed in the macrometastases do exhibit partial EMT program in this system. We revised our discussion to further highlight the potential limitations.

The new staining for E-cadherin in primary tumors (Suppl. Figure S4C) shows that tdTomato+ cells have lost E-cadherin expression, thus technically validating the actual lineage tracing method, but it does not add any insights on the distinction between CK19- cancer cells and stromal cells.

- This is not what our data show. In Suppl. Figure S4C, E-cadherin expression is noted in cancer cells with an EMT program, similar to CK19, albeit, with some down regulation that is expected.

The authors still categorize the few tdTomato+ cancer cells that are found in micrometastasis as having undergone a partial EMT. They now provide more discussion on the definition of a partial EMT in the Discussion section, yet they do not really show hard data to demonstrate a partial EMT in the "EMT-like" cells found in micrometastases.

- Our approach to define these cells are in line with similar strategies used in others studies that propose the identity of cell with 'partial' EMT program.

The authors now include a discussion of the report by Krebs et al. (2018) which shows reduced metastasis by genetic ablation of Zeb1 expression in the KPC mouse model of PDAC. However, it is stunning to see that they interpret the results by Krebs et al. (2018) as Zeb1 being "dispensable" for metastasis formation. The authors should not ignore the redundancy between the various EMT core transcription factors. Moreover, the experiments reported by Krebs et al. analyze the functional contributions of Zeb1 to tumor progression and metastasis and not explicitly to EMT.

- In the study titled: "The EMT-activator Zeb1 is a key factor for cell plasticity and promotes metastasis in pancreatic cancer", the authors acknowledge Zeb1 as an EMT activator, in agreement with many studies claiming this transcription factor is critical for EMT program. The reviewer is asked to review the data in Figure 1F in Krebs et al. (NCB, 2017). There the authors show that 13.8% of KPC mice with complete loss of Zeb1 exhibit metastasis, when compared to 36.5% of KPC control. As such, ~50% of the mice preserve metastatic potential despite loss of Zeb1, indicating that Zeb1 is dispensable for metastasis. The reviewer is correct in stating that redundancy among EMT inducing transcription factors must be considered when suppression of partial EMT is not achieved when one of them is genetically deleted.

Corresponding Author Name: Raghu Kalluri

Manuscript Number: EMM-2018-09085